# Acquisition of innate odor preference depends on spontaneous and experiential activities during critical period

Qiang Qiu[1], Yunming Wu[1], Limei Ma[1], Wenjing Xu[1], Max Hills Jr[1], Vivekanandan Ramalingam[1,2], C Ron Yu[1,2,3]*

[1]Stowers Institute for Medical Research, Kansas City, United States; [2]Interdisciplinary Graduate Program in Biomedical Sciences, University of Kansas Medical Center, Kansas City, United States; [3]Department of Anatomy and Cell Biology, University of Kansas Medical Center, Kansas City, United States

**Abstract** Animals possess an inborn ability to recognize certain odors to avoid predators, seek food, and find mates. Innate odor preference is thought to be genetically hardwired. Here we report that acquisition of innate odor recognition requires spontaneous neural activity and is influenced by sensory experience during early postnatal development. Genetic silencing of mouse olfactory sensory neurons during the critical period has little impact on odor sensitivity, discrimination, and recognition later in life. However, it abolishes innate odor preference and alters the patterns of activation in brain centers. Exposure to innately recognized odors during the critical period abolishes the associated valence in adulthood in an odor-specific manner. The changes are associated with broadened projection of olfactory sensory neurons and expression of axon guidance molecules. Thus, a delicate balance of neural activity is needed during the critical period in establishing innate odor preference and convergent axon input is required to encode innate odor valence.

*For correspondence:
cry@stowers.org

## Introduction

Behavioral characteristics are often described as either acquired or innate. While most environmental stimuli do not carry obvious ethological values, animals can develop characteristic responses through associative learning and assign valence to individual stimuli. Animals also react innately to some stimuli with instinctive responses that are thought to be pre-programmed in the neural circuits. These stereotypic responses likely have evolved to deal with stimuli in the animal's immediate environment that carry information about the inherent values of the signals. Odor-based predator avoidance, food seeking, pheromone-induced territorial aggression, and mating are examples of such innate responses. The exhibition of stereotypical responses to these stimuli often has an onset associated with developmental stages and does not require any prior experience. In most sensory systems, circuit modification by sensory experience underlies behavioral adaptation. In the mammalian brain, circuit connections are sensitive to sensory deprivation during early postnatal development in a time window defined as the critical period (*Hensch, 2005*; *Hubel and Wiesel, 1970*). Learning further modifies circuit connections to allow flexible assignment of valence to specific sensory stimuli. Circuit motifs such as divergent paths and opposing components allow adaptive changes that are thought to mediate these flexible assignments (*Tye, 2018*). On the other hand, innate behaviors are defined by stereotypical responses to the stimuli without prior experiences or associative learning, suggesting that neural circuits processing the inherent valence of stimuli are genetically hardwired to channel sensory information directly to the motor or endocrine output.

The olfactory system mediates both learned and innate responses to odor stimuli. A repertoire of ~1000 functional odorant receptors (ORs), 15 TAARs, and a family of MS4A receptors detect volatile odors (*Buck and Axel, 1991*; *Greer et al., 2016*; *Liberles and Buck, 2006*; *Pacifico et al., 2012*). Olfactory sensory neurons (OSNs) expressing the same OR converge their axons into a few glomeruli stereotypically positioned in the olfactory bulb forming a topographic map to encode odor quality (*Axel, 2005*). Odors activate disparate sets of glomeruli and the patterns are transformed by the mitral/tufted cells in the olfactory bulb before the information is passed to the olfactory cortices. The patterns of activity in the cortical areas can be learned through association with other stimuli. Mice are also intrinsically attracted by food odors and conspecific urine but avoid odors from decomposing flesh or predators. The dorsal area of the olfactory bulb is shown to be required for innate responses to aversive odors; other studies have shown that a few ORs are both sufficient and necessary to trigger innate responses to specific odors (*Dewan et al., 2013*; *Ishii et al., 2017*; *Kobayakawa et al., 2007*; *Pérez-Gómez et al., 2015*; *Zhang et al., 2013*). However, recent studies have indicated that innately recognized odors are encoded by more flexible ensemble activities (*Iurilli and Datta, 2017*; *Qiu et al., 2021*).

It is not known whether neural circuits that underlie innate responses are influenced by sensory experience or subject to changes of neural activity during development. In postnatal development of the olfactory system, OSNs fire spontaneous action potentials originating from ligand-independent activities (*Nakashima et al., 2013*; *Yu et al., 2004*). The receptor activities drive specific temporal patterns of action potential firing (*Nakashima et al., 2019*). Spontaneous activity influences the development of the olfactory glomerular map during the critical period (*Ma et al., 2014*; *Tsai and Barnea, 2014*; *Wu et al., 2018*), which persists into adulthood. Interestingly, the critical period in development coincides with a form of learning in young animals. Early olfactory experiences are critical for filial learning and attachment to the maternal odor environment in ewes and rodents, nest recognition in songbirds, and kin recognition in fish and rodents (*Caspers et al., 2013*; *Gerlach et al., 2008*; *Hofer et al., 1976*; *Polan and Hofer, 1999*; *Porter et al., 1978*). Notably, exposure-based learning during early infancy via olfactory imprinting differs from odor learning in adulthood. Pairing odor with an unconditioned painful stimulus produces attachment in pups instead of aversion as found in adults (*Sullivan et al., 2000*). Given the importance of the critical period in shaping olfactory experience, we wonder whether changes caused by levels of spontaneous activity or odor experience have any impact on innate odor-evoked behaviors.

## Results

### Spontaneous activity of the OSNs is required for innate odor recognition

In previous studies, we developed a genetic approach to silence spontaneous activity during development (*Ma et al., 2014*; *Yu et al., 2004*; *Figure 1A–i*). Ectopic expression of the *Kcnj2* (*Kir2.1*) channel gene in the *Omp-IRES-tTA/tetO-Kcnj2-IRES-taulacZ* compound heterozygotic mice suppressed spontaneous activities (*Yu et al., 2004*). Feeding the mice with doxycycline (DOX) suppressed *Kcnj2* expression (*Ma et al., 2014*). We restricted ectopic *Kcnj2* expression to the pre-weaning period through DOX administration (*Figure 1A–ii*), thereby providing an opportunity to test whether spontaneous activity during early development is required for innate odor preference.

We tested whether suppression of neural activity during early development has a long-lasting effect on neuronal physiology. We performed electrophysiological recordings from the DOX-treated mice (referred to as the Kir2.1-off mice) in adulthood. OSNs expressing various ORs exhibit various firing patterns. Random sampling from a heterogeneous population of OSNs would not provide a precise measure of spontaneous firing. We, therefore, used cell-attached patch clamp to record spontaneous activity from the dendritic knobs of sensory neurons expressing the *Olfr160* (*M72)* gene in the *Olfr160-IRES-tauGFP* line (*Feinstein and Mombaerts, 2004*; *Figure 1B–i*). Neurons in Kir2.1-off mice displayed similar inter-spike intervals and spontaneous firing rates compared to those in the control mice (*Figure 1B–ii–iv*). We next conducted electro-olfactogram (EOG) recordings to measure the field potential of odor-evoked responses. We found that there was no discernible difference in the amplitude and time course of odor-evoked responses between control and Kir2.1-off mice (*Figure 1C*). Thus, DOX treatment restored neural activity in the OSNs of Kir2.1-off animals.

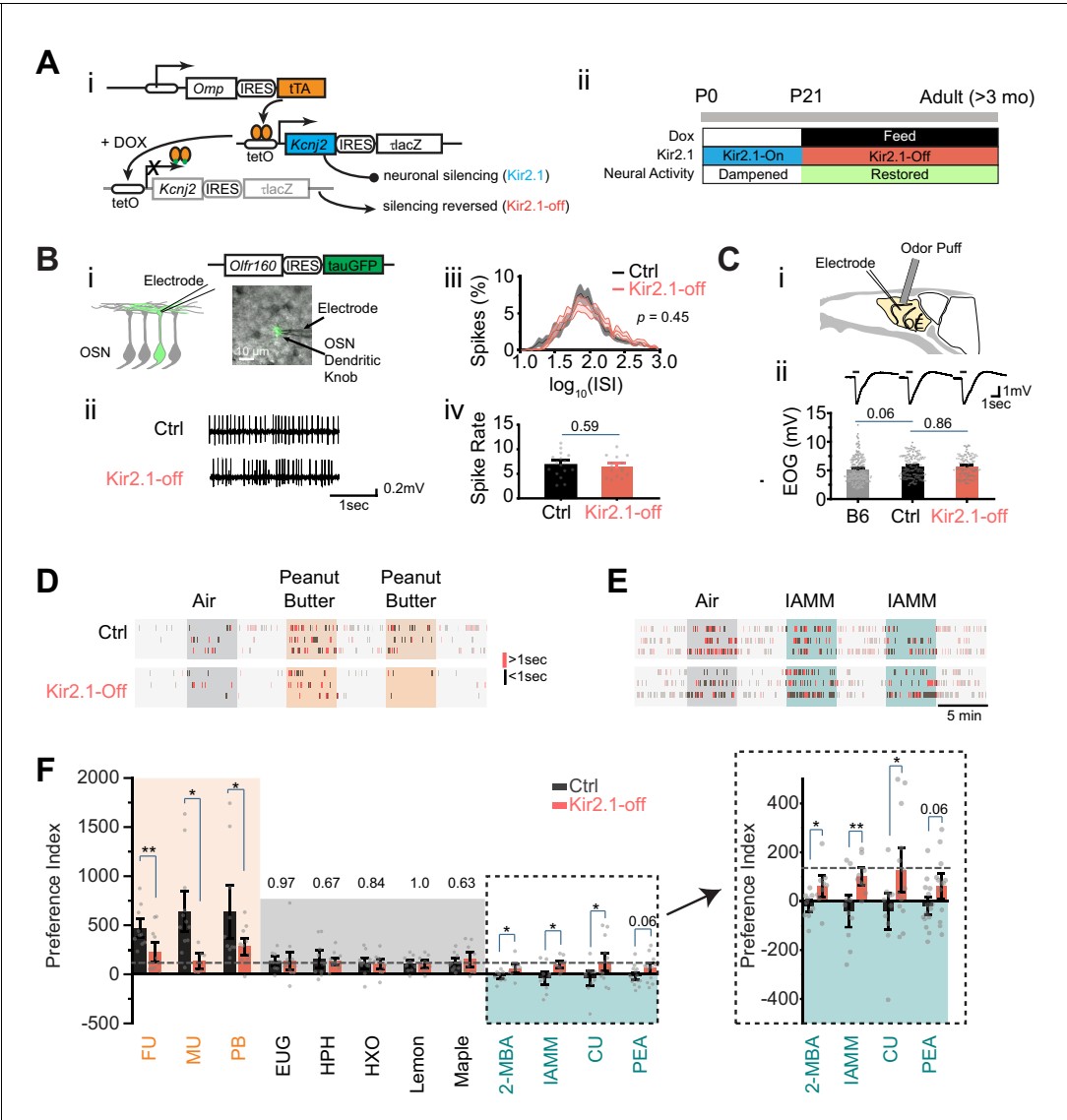

**Figure 1.** Suppressing spontaneous activity during early development alters odor preference in adulthood. (**A**) i. Illustration of the inducible transgene expression in the *Omp-IRES-tTA/tetO-Kcnj2-IRES-tauLacZ* (*Kir2.1*) mice. ii. Schematic illustration of genetic silencing ol factory sensory neurons (OSNs) during early postnatal period. (**B**) i. Dendritic recording of an OSN in *Olfr160-IRES-tauGFP* (M72-GFP) animal. Superimposed bright field and fluorescent images show a recording pipette attached to the dendritic knob of an M72-GFP OSN. ii. Raster plots showing examples of spontaneous spiking patterns of OSNs from control and Kir2.1-off mice. iii. Distribution of inter-spike interval (ISI) in control (black, neuron n = 16) and Kir2.1-off (red, n = 14) animals. Spike count: Ctrl, 8187; Kir2.1-off, 5076. The ISIs follow lognormal distributions. One-way ANOVA test on average log (ISI) was performed, p=0.45. iv. Bar plot of average firing frequencies of M72 OSNs in control (black) and Kir2.1-off (red) mice. (**C**) i. Illustration of electro-olfactogram (EOG) recording. ii. Response sample traces (top) and average amplitude (bottom) of C57Bl/6 (B6, n = 189), littermate control (Ctrl, n = 132), and Kir2.1-off mice (n = 104). (**D and E**) Raster plots of odor port investigation of air, peanut butter (PB, **D**), and isoamylamine (IAMM, **E**). Each tick represents an investigation event. Investigations longer and shorter than 1 s are marked by red and black ticks, respectively. Preference index is calculated as the average difference of port investigations between the first two odor epochs and the last air epoch. (**F**) Bar graph showing preference indices for control (black) and Kir2.1-off (red) mice. Shaded areas indicate innate preference to animals: orange indicates attraction; blue indicates aversion; gray indicates neutral. The gray dashed line indicates the average of preference indices to the five neutral odors. Inlet: enlarged graph showing preference indices to aversive odors. The following odorants were used. Neutral odors, eugenol (EUG, Ctrl n = 9, Kir2.1-off n = 12), heptanal (HPH, Ctrl n = 9, Kir2.1-off n = 9), 2-hexanone (HXO, Ctrl n = 10, Kir2.1-off n = 10), lemon (Ctrl n = 9, Kir2.1-off n = 9), and maple (Ctrl n = 10, Kir2.1-off n = 9); attractive odors, PB (Ctrl n = 10, Kir2.1-off n = 11), female urine (FU, Ctrl n = 10, Kir2.1-off n = 11), and male urine (MU, Ctrl n = 12, Kir2.1-off n = 6); aversive odors, 2-MBA (Ctrl n = 10, Kir2.1-off n = 9), IAMM (Ctrl n = 10, Kir2.1-off n = 9), coyote urine (CU, Ctrl n = 10, Kir2.1-off n = 13), and 2-phenylethylamine (PEA, Ctrl n = 15, Kir2.1-off n = 13). All bar graph data are shown in mean ± SEM. Individual data points are shown as gray dots. One-way ANOVA performed. *, p<0.05; **, p<0.01; ***, p<0.01.

The online version of this article includes the following source data for figure 1:

**Source data 1.** Quantification of neuronal activity and odor preference.

We next tested innate odor preference in the Kir2.1-off mice using the PROBES system (*Qiu et al., 2014*). This assay exploited the animal's natural tendency to investigate a stimulus. We measured the duration of odor source investigation and compared it to the no odor period. The level of investigation indicated odor preference (*Qiu et al., 2014*). Control mice exhibited increased frequency and duration in sniffing the odor port during the presentation of peanut butter (PB) odor or mouse urine (*Figure 1D*). The increase in investigation sustained into the second and third presentations. In contrast, when odors aversive to the animals were presented the first time, control mice approached the odor port at a level similar to that of air control (*Figure 1E*). However, at the second presentation, we observed a marked reduction in the investigation of the odor port. It was likely that the investigative behavior during the first odor presentation was driven by two conflicting drives: risk assessment (approaching) and avoidance. To have a consistent measurement of odor preference we combined the changes at the first and second odor epochs into a single index to measure the innate behavioral preference (see Materials and methods) (*Qiu et al., 2021*).

From this assay it was clear that odors of PB and mouse urine elicited strong attraction when compared with monomolecular odors that were generally considered neutral (*Figure 1F*). In contrast, the Kir2.1-off mice did not exhibit attraction to food and conspecific urine odors (*Figure 1D, F*). The level of investigation of the odor port was comparable to those of the neutral odors and did not sustain after the first presentation. Control mice also showed a much lower preference towards odors considered as aversive, including coyote urine (CU), 2-phenylethylamine (PEA), 2-methylbutyric acid (2-MBA), and isoamylamine (IAMM). The Kir2.1-off mice, on the contrary, continued to investigate the aversive odors at a level similar to that for the neutral odors (*Figure 1E and F*). These results indicated that silencing OSNs during early postnatal development impaired the ability of the mice to recognize the valence of the ethologically relevant odors.

## Exposure during critical period alters innate odor recognition

The loss of innate odor preference following neuronal silencing during postnatal development was surprising. It suggested that the neural circuits processing innate valence were plastic during development. We hypothesized that this developmental plasticity would allow the early olfactory experience to influence odor-evoked innate responses later in life. We chose PEA as an aversive odor to test this hypothesis since it specifically activated the TAAR4 receptor, which is necessary for PEA-induced aversion (*Dewan et al., 2013*; *Zhang et al., 2013*).

We put an odor tube in the cages where the pups were raised during the first two postnatal weeks (*Figure 2A*). The concentration of PEA in the cage was comparable with that used for behavioral tests (*Figure 2—figure supplement 1*). Following another 7 weeks without exposure, we examined the animals' response to PEA (*Figure 2A*). To test whether the influence of PEA was restricted to a specific time window, we exposed another group of animals to PEA between postnatal days 24 and 38, followed by testing at 9 weeks of age (*Figure 2B*). To determine the odor specificity of this change in responses, we also tested behavioral response to IAMM, which elicited innate aversion but was not exposed to the animals during these periods.

Compared to the control mice that showed aversion to PEA, the P0–P14-treated animals did not exhibit aversion to PEA (*Figure 2C*). These mice, however, still exhibited aversive response to IAMM (*Figure 2D*). The timing of PEA exposure also had a profound impact on odor aversion in adults. In contrast to the early postnatally exposed mice, animals that had late exposure to PEA exhibited aversion to both PEA and IAMM (*Figure 2C–D*). Thus, the effect we have observed for the P0–P14 animals was not due to odor adaptation from persistent exposure but reflected a profound change in how PEA was perceived.

How does early PEA exposure change valence? One possibility is that the innate association between odor identity and valence is perturbed. This would be similar to the results observed in the neuronal silencing experiments. Another possibility is that through exposure, the aversive odor also acquires positive valence, as suggested by previous studies (*Sullivan et al., 2000*). In the latter scenario, pre-exposure to an attractive odor will not alter the positive valence. We, therefore, tested the perinatal exposure to attractive odors. We used TMA, which has been characterized as attractive in past studies, and PB odor. In both cases, perinatal exposure extinguished innate attraction in the adults for the odor exposed, but not the un-exposed odor (*Figure 2E and F*). Taken together, these results showed that odor exposure disrupted odor association with valence rather than counterbalancing it.

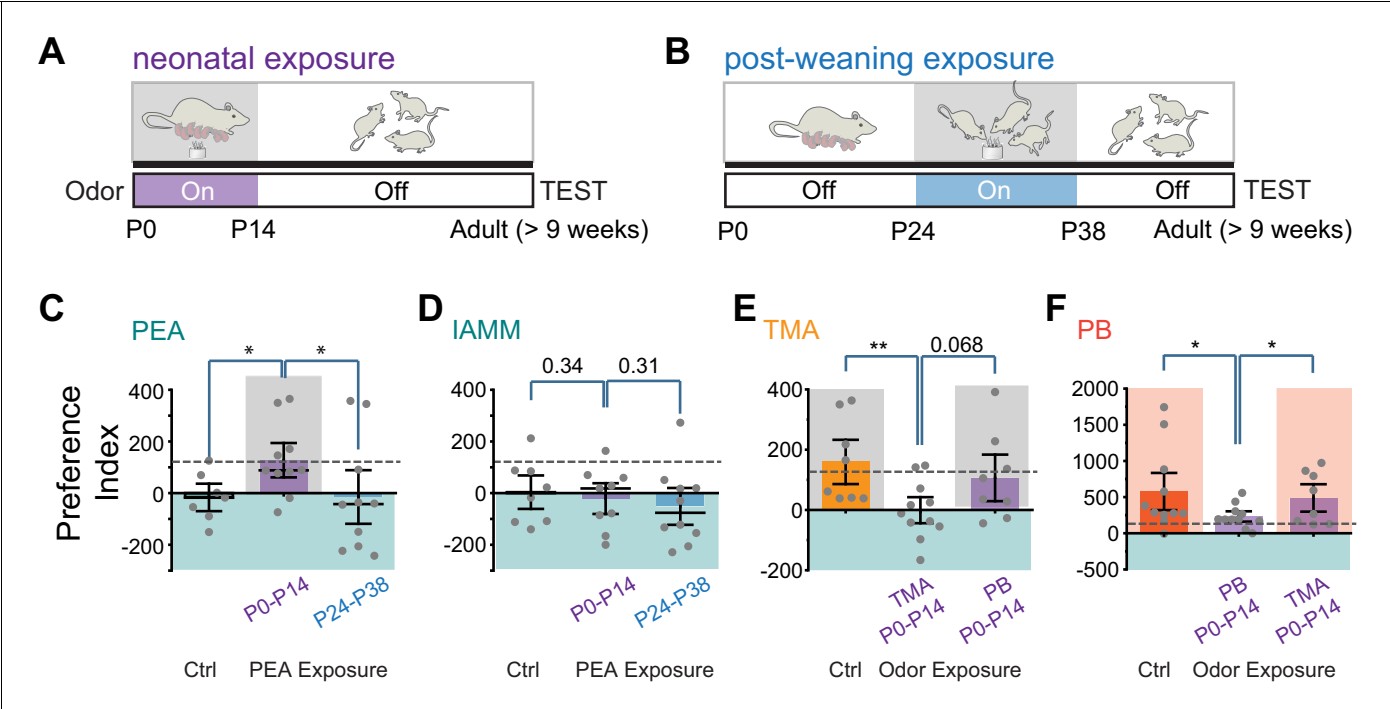

**Figure 2.** Early postnatal exposure to odors alters innate odor recognition in adulthood. (**A and B**) Schematic illustration of the experimental paradigms. CD-1 pups were exposed to 2-phenylethylamine (PEA) between P0 and P14 (**A**) or between P24 and P38 (**B**). The animals were tested at the adult stage. Shaded areas indicate PEA exposure. (**C and D**) Bar plots of preference indices to PEA (**C**, Ctrl n = 7, P0–P14 PEA exposed n = 10, P24–P38 PEA exposed n = 10) and IAMM (**D**, Ctrl n = 8, P0–P14 PEA exposed n = 9, P24–P38 PEA exposed n = 10). (**E and F**) Bar plots of preference indices to TMA (**E**, Ctrl n = 8, P0–P14 TMA exposed n = 11, P0–P14 peanut butter [PB] exposed n = 8) and PB (**F**, Ctrl n = 11, P0–P14 PB exposed n = 11, P0–P14 TMA exposed n = 8). The gray dashed line indicates the average preference to neutral odors as in *Figure 1F*. One-way ANOVA performed. *, p<0.05. The online version of this article includes the following source data and figure supplement(s) for figure 2:

**Source data 1.** Quantification of odor preference after early postnatal exposure.
**Figure supplement 1.** Measuring odor concentration in home cage.
**Figure supplement 1—source data 1.** Numerical data of PID measurement of odor signal.

## Neural activities influence axon projection pattern of TAAR4 expressing neurons

How does temporary suppression of action potentials, or the exposure to aversive odor during early postnatal development abolish innate odor preference? Odor valence could be conveyed by specific parts of the olfactory bulb. For example, the dorsal bulb was shown to be required for conveying aversive information of aversive odors (*Kobayakawa et al., 2007*). Alternatively, but not mutually exclusively, a highly specific link between the ORs, a set of mitral/tufted cells, and their downstream targets could be established through genetically specified hardwiring regardless of the spatial locations where the receptor neurons project. In previous studies, we have shown that suppression of neural activity during early development caused the olfactory axons expressing the same receptor to innervate multiple glomeruli (*Ma et al., 2014*; *Nakashima et al., 2013*; *Yu et al., 2004*; *Zhao et al., 2013*). We also found that the OSNs projected to the same dorsal–ventral and anterior–posterior position of the olfactory bulb where wild-type axons innervated, suggesting that the general topology of OSN innervation remained intact and the OR gene expression was maintained. However, it was not known whether the broadened projection patterns also applied to receptor neurons that transmit identity information of innately recognized odors.

We, therefore, examined whether the OSNs detecting an aversive odor maintain their projection specificity. We performed immunofluorescence staining using antibodies against TAAR4 (*Johnson et al., 2012*), which detected PEA and was required for aversive response to PEA (*Pacifico et al., 2012*). We found that in the Kir2.1-off mice the axons expressing TAAR4 projected

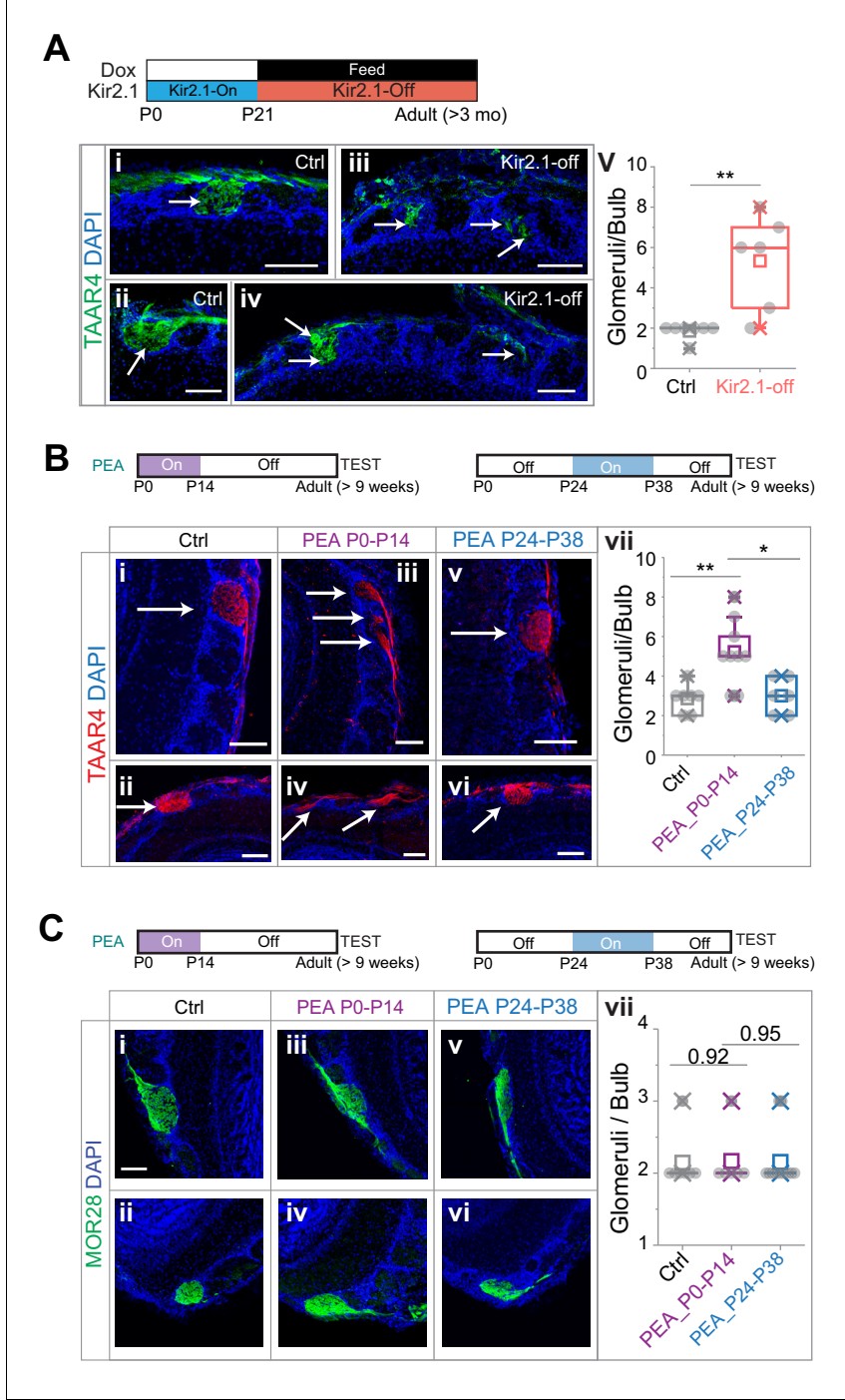

**Figure 3.** Altered projection patterns of TAAR4 expressing olfactory sensory neuron (OSN) axons in Kir2.1-off and 2-phenylethylamine (PEA) treated mice. (**A**) Immunofluorescent staining of TAAR4 receptor in olfactory bulb sections from control animals (i and ii, n = 7) and Kir2.1-off animals (iii and iv, n = 6). Green, TAAR4 signal. Blue, DAPI. v. Quantification of TAAR4+ glomeruli each OB. Box plot edges indicate the first and third quartiles of the data, while whiskers indicate 1.5 interquartile range. (**B**) TAAR4 staining of olfactory bulb sections for control (i and ii, n = 7), P0–P14 PEA exposed (iii and iv, n = 9) and P24–P38 exposed (v and vi, n = 6) CD-1 animals. Red, TAAR4. Blue: DAPI. vii. Quantification of TAAR4+ glomeruli for each OB. (**C**) Confocal images of MOR28 receptor immunofluorescent staining in olfactory bulb sections for control (i and ii, n = 7), P0–P14 PEA exposed (iii and iv, n = 6) and P24–P38 PEA exposed (v and vi, n = 13) animals. Green, MOR28. Blue: DAPI. vii. Quantification of MOR28+ glomeruli for each OB. One-way ANOVA result is shown. **, p<0.01. Arrows point to each glomerulus. Scale bars, 100 μm.

*Figure 3 continued on next page*

Figure 3 continued
The online version of this article includes the following source data for figure 3:
**Source data 1.** Quantification of axon projection changes induced by Kir2.1 expression and PEA exposure.

to more glomeruli than the wild-type mice (*Figure 3A*; 1.86 ± 0.38 and 5.33 ± 2.34 glomeruli per dorsal OB for control and Kir2.1-off, respectively. One-way ANOVA, p=0.0025). Nevertheless, the glomeruli receiving TAAR4 projection were located in the dorsal and medial olfactory bulb, in the same general region as the wild-type TAAR4 glomeruli.

We next performed the same experiments on mice that were exposed to PEA during the first two postnatal weeks. We found that PEA exposure also led to divergent projection of TAAR4 expressing axons (*Figure 3B*). TAAR4 axons were found on average 5.22 ± 1.64 glomeruli per OB, significantly more than in the control mice (2.53 ± 0.69; one-way ANOVA applied, p=0.0032). Moreover, the number of TAAR4 glomeruli in mice exposed to PEA between P24 and P38 was indistinguishable from the controls (3.0 ± 0.89, p=0.75).

We also stained the sections using antibodies against the receptor MOR28, which did not respond to PEA (*Barnea et al., 2004*). The number of glomeruli innervated by MOR28 OSNs was not affected by the PEA exposure (*Figure 3C*, 2.14 ± 0.38, 2.17 ± 0.41 and 2.09 ± 0.30 glomeruli per OB for control, P0–P14 PEA exposure and P24–P38 PEA exposure respectively, p=0.92 and 0.95). As the TAAR4 neurons still innervated the dorsal olfactory bulb, these observations suggested that the activation of the dorsal bulb was not sufficient to signal aversion.

## Spontaneous neural activity during early development is not required for odor detection

Convergence of OSNs expressing the same receptor is a signature of the olfactory system and has been suggested to play important roles in odor coding, signal amplification, and odor discrimination (*Chen and Shepherd, 2005*; *Cleland and Linster, 2005*; *Su et al., 2009*; *van Drongelen et al., 1978*; *Wilson and Mainen, 2006*; *Zou et al., 2009*). We wondered whether the change in OSN projection patterns altered the ability of the mice to detect or discriminate odors, which could offer a trivial explanation to the change in the innate odor preference.

In behavioral experiments, the lack of odor source investigation could be due to aversion or a lack of detection. We, therefore, used neutral odors to test odor detection and discrimination in the Kir2.1-off mice. We evaluated detection of low concentration of amyl acetate (AA) and 2-heptanone (HPO) using a dishabituation assay that provided a sensitive readout of odor detection threshold in naïve animals (*Qiu et al., 2014*). Control and Kir2.1-off mice exhibited similar increases in investigation of the odor port with increasing concentration of odors (*Figure 4A–i* and *Figure 4—figure supplement 1A*). The thresholds of detecting AA were $1.90 \times 10^{-6}$ and $1.56 \times 10^{-6}$ v/v for control and Kir2.1-off mice, respectively (*Figure 4A–i*). We used p-values to identify the first concentration to elicit a noticeable increase in odor port investigation, and found that both Kir2.1-off and control mice have significantly increased investigation at $1 \times 10^{-6}$ v/v for AA and HPO (*Figure 4A–ii* and *Figure 4—figure supplement 1B*).

We performed additional evaluation of detection thresholds using the Go/No Go task (*Qiu et al., 2014*). Water-restricted animals were trained to associate an odor with water reward and then tested with lowered concentrations of the trained odor. From this test, we determined that the thresholds of detecting AA were $6.64 \times 10^{-6}$ and $7.88 \times 10^{-6}$ v/v for control and Kir2.1-off mice, respectively (*Figure 4B*). The threshold values were slightly higher than those obtained from dishabituation assays, but at the same order of magnitude. This was consistent with our previous findings and might reflect a requirement for the mice to decide whether or not to lick the water spout (*Qiu et al., 2014*). To examine whether the loss of aversion was due to compromised detection of these odors, we tested the detection of PEA in Kir2.1-off and PEA-exposed animals. Because PEA is aversive, we only tested the lowest concentrations near threshold. Kir2.1-off mice and PEA-exposed mice could detect PEA at $10^{-13}$ M and $10^{-12}$ M, respectively (*Figure 4—figure supplement 1C and D*). These thresholds were comparable with those exhibited by the control mice and were consistent with previously reported values (*Dewan et al., 2018*).

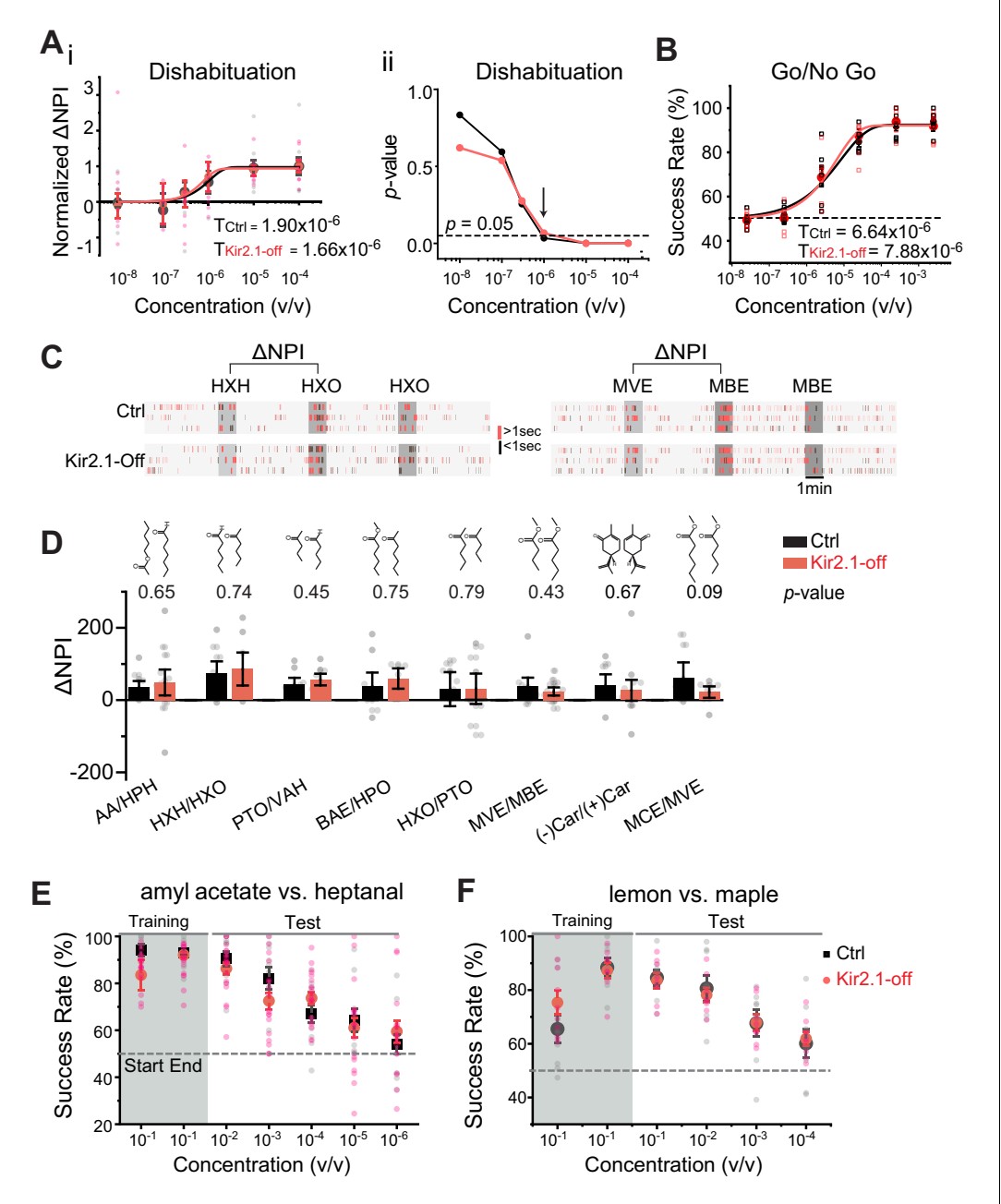

**Figure 4.** Suppressing spontaneous activity during early development does not impact odor detection, discrimination, and association. (**A**) Detection threshold of amyl acetate (AA) is determined by dishabituation assay. i. Mean ΔNPI values for AA presented at different concentrations ($10^{-8}$ to $10^{-4}$ v/v) for control (black, n = 9) and Kir2.1-off (red, n = 10) mice. Data are fitted with Weibull psychometric function with threshold values (T) shown. ii. p-values of ΔNPI at different odor concentrations from (i). Arrow indicates the concentration at which p-value is at 0.05. Dashed line indicates where the p-value is 0.05. (**B**) Success rates in Go/No Go test with increasing odor concentrations of AA (Ctrl, n = 9; Kir2.1-off, n = 9). Data are fitted with a Weibull psychometric function. Threshold values (T) calculated from the fitting are indicated. Dashed line in indicates success rate at chance level of 50%. (**C**) Cross habituation test. Raster plots of odor port investigation by control (Ctrl) and Kir2.1-off mice. Animals are habituated with hexanal (HXH, left) or methyl valerate (MVE, right) before exposed to 2-hexanone (HXO, left) or methyl butyrate (MBE, right) respectively. Odor deliveries are marked by gray boxes. Only the last habituation and first two novel stimulation periods are shown. (**D**) Bar plot of ΔNPI (mean ± SEM) in Ctrl (black) and Kir2.1-off (red) for discrimination of different odor pairs. Chemical structures of the odorants are shown. Numbers above the bars indicate the p-values between the scores obtained from Ctrl and Kir2.1-off. Animals

*Figure 4 continued on next page*

*Figure 4 continued*

used: AA/HPH, Ctrl n = 10, Kir2.1-off n = 14; HXH/HXO, Ctrl n = 9, Kir2.1-off n = 8; PTO/VAH, Ctrl n = 9, Kir2.1-off n = 9; BAE/HXO, Ctrl n = 9, Kir2.1-off n = 6; HXO/PTO, Ctrl n = 9, Kir2.1-off n = 10; MVE/MBE, Ctrl n = 10, Kir2.1-off n = 14; (−)Car/(+)Car, Ctrl n = 8, Kir2.1-off n = 14;, MCE/MVE, Ctrl n = 7, Kir2.1-off n = 8. (E and F) Two-choice odor discrimination assay. Scatter plots show the success rate for Ctrl (black, n = 4) and Kir2.1-off (red, n = 5) mice in discriminating AA versus heptanal (E), and lemon versus maple (Ctrl, n = 8; Kir2.1-off, n = 8) (F) at decreasing concentrations.

The online version of this article includes the following source data and figure supplement(s) for figure 4:

**Source data 1.** Quantification of odor detection and discrimination.
**Figure supplement 1.** Odor detection at low concentration.
**Figure supplement 1—source data 1.** Quantification of odor detection and discrimination.

---

Taken together, neuronal silencing during early postnatal development did not alter odor sensitivity in mice. Moreover, the animals were able to learn the association between odors and reward during the Go/No Go training, suggesting that learned valence association was not affected either. These results ruled out the possibility that the loss of innate aversion was caused by diminished ability to detect the odors.

## Kir2.1-off mice exhibit normal odor discrimination and learning

We next examined whether Kir2.1-off mice lost their ability to discriminate odorants. Specifically, we aimed to test the animals' innate ability to discriminate odors without prior exposure. To achieve this goal, we adopted the cross habituation assay using PROBES (*Qiu et al., 2014*). This assay exploited the animal's natural tendency toward habituation following repeated exposures to the same stimulus carrying no apparent value and novelty seeking behavior when a new stimulus was presented after habituation (*Figure 4C*). We measured the duration of odor source investigation and normalized the values to that during the control (no odor) period to derive normalized port investigation (NPI) values (*Qiu et al., 2014*) (also see Materials and methods). These NPI values allowed us to compare among animals with varied individual behavior readout. We then used ΔNPI, the difference between NPI values for the novel and habituated odors, to measure discrimination between the odors. For eight pairs of mono-molecular odorants tested, the Kir2.1-off mice could discriminate them all (*Figure 4D*). The scores for the Kir2.1-off mice were higher than those of the controls for some odors but not the others. None of the differences was statistically significant. Thus, the Kir2.1-off mice were not deficient in discriminating odors under naïve conditions. We also tested discrimination between PEA and a neutral odor, eugenol, in the Kir2.1-off and PEA exposed mice. In both experiments, the animals were able to discriminate between the two odors (*Figure 4—figure supplement 1E and F*).

Finally, we tested the ability of odor discrimination and associative learning in Kir2.1-off mice with the reinforced two-choice assay. In this assay, water-restricted mice learned to distinguish a pair of odors and associate them with the spatial location of water ports in order to receive water reward (*Qiu et al., 2014*; *Uchida and Mainen, 2003*). The assay not only required the animals to discriminate odors, but also assigned valence to the odors. Both control and Kir2.1-off mice achieved ~90% success rate after training to discriminate a pair of mono-molecular odors, AA, and heptanal (*Figure 4E*), and between a pair of complex odors, lemon and maple (*Figure 4F*). In tests using lower concentrations of odors, we observed a similar decline in accuracy between the two groups (*Figure 4E and F*). These results indicated that the Kir2.1-off mice not only could discriminate odors, but also could recognize individual odors and associate them with the correct water ports. Importantly, the Kir2.1-off mice were able to assign valence to odors during learning. Taken together, these results showed that the Kir2.1-off had no deficiency in odor discrimination, recognition, and valence association.

## Mismatch between OSN and mitral/tufted cells by altered neural activities

We next examined the structural basis of altered odor perception. In wild-type mice, individual olfactory glomeruli are innervated by OSN axons expressing the same OR and by the primary dendrites

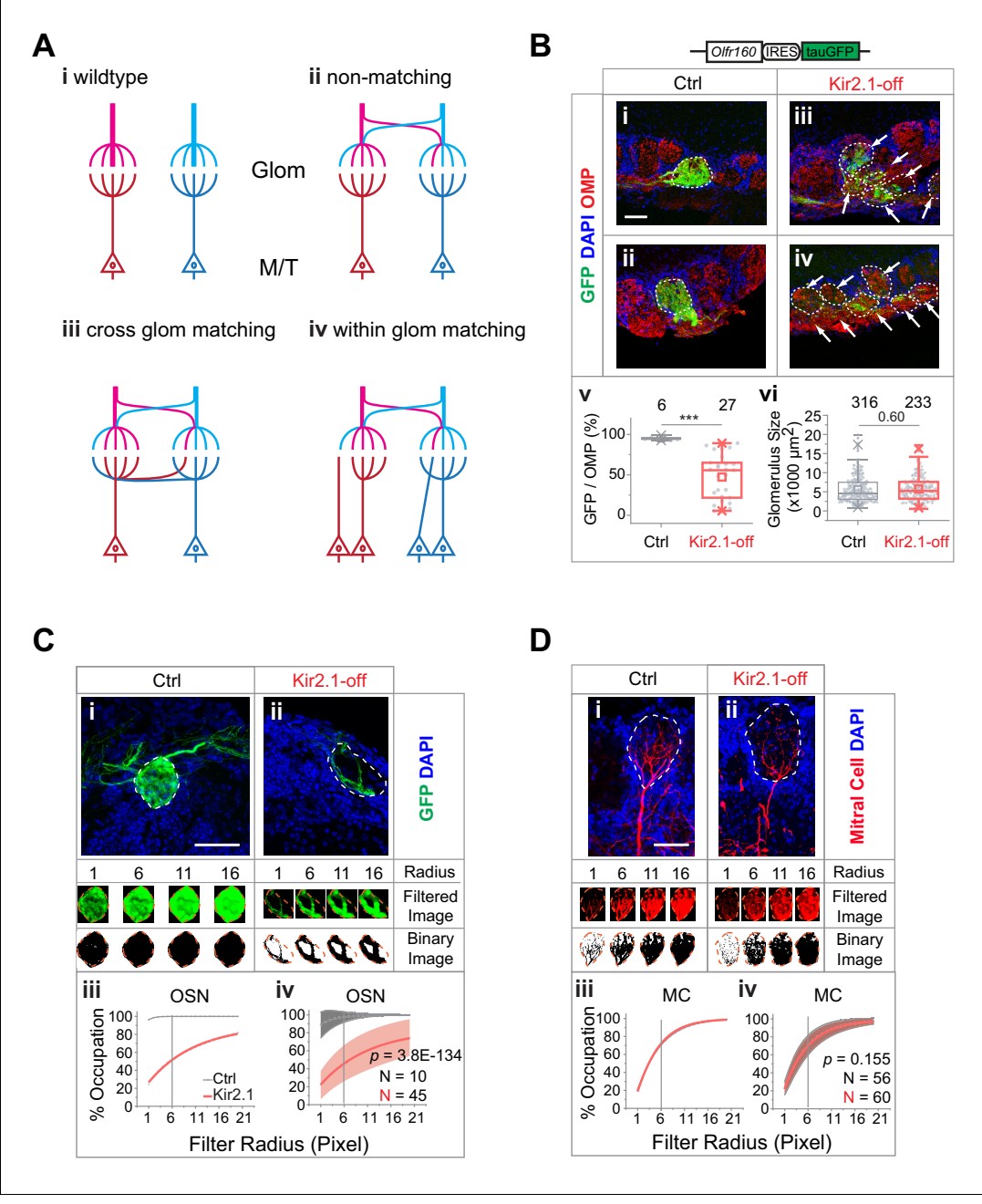

**Figure 5.** Mismatch between olfactory sensory neuron (OSN) axons and M/T cell dendrites in Kir2.1-off mice. (**A**) Models of connectivity between OSN axons and mitral/tufted cell dendrites. i, In the wild type, each glomerulus receives input from axons expressing the same odorant receptors (ORs; color coded), which are connected to the same primary dendrites of the mitral/tufted cells. ii, Divergent projection of OSN axons does not influence dendritic innervation such that the primary dendrites receive input from OSNs expressing different ORs. iii, Axon-dendritic connection is made through cross-glomerular matching such that the same dendrite is always innervated by OSNs expressing the same OR. iv, Matching axon-dendritic connection is restricted only in the glomeruli. (**B**) Confocal images of olfactory bulb sections from mice carrying homozygotic *Olfr160-IRES-tauGFP* (M72) allele. Immunofluorescent signals show GFP (green), DAPI (blue), and OMP (red) in control (i and ii) and Kir2.1-off (iii and iv) mice. Dotted circles circumscribe the glomeruli containing green fibers. Glomeruli are identified based on the density of periglomerular cell nuclei. Arrows point to fibers in the glomerular layer that are not counted toward quantification. Scale bar, 50 µm. v. Quantification of the overlap between OMP signals and GFP signal in individual glomeruli in the control and Kir2.1-off mice. vi. Quantification of the glomerular size (diameter) in control and Kir2.1-off mice (N = 3 animals each). Numbers in the panel indicate the total glomeruli counted. (**C**) Glomerular

*Figure 5 continued on next page*

*Figure 5 continued*

occupancy by OSN axons. Immunofluorescent staining in Ctrl (i) and Kir2.1-off (ii) mice to label *Olfr160-IRES-tauGFP* (M72) axons (green). Spatial filters with 1-, 6-, 11-, and 16-pixel radii were applied to obtain filtered images. After applying a binary threshold, any pixel that contains fluorescent signal is shown in black. Scale bar, 50 µm. Percentages of glomerular occupancy calculated as a function of filter radius are plotted for individual images (iii) for images shown in (i) and for all data (iv). Data are shown in mean (dashed line) ± standard deviation (shaded area). Two-way ANOVA applied for comparing Kir2.1-off and control, p=3.8E-134. (D) Same as C but for mitral cell dendrites (BDA, red). Two-way ANOVA applied for comparing Kir2.1-off and control, p=0.155. Note that black and red lines almost overlap in iii and iv. Scale bar, 50 µm.

The online version of this article includes the following source data and figure supplement(s) for figure 5:

**Source data 1.** Quantification of axon and dendritic innervation of olfactory glomeruli.

**Figure supplement 1.** Ol factory sensory neuron (OSN) projection patterns in the Kir2.1-off animals.

**Figure supplement 1—source data 1.** Quantification of OSN axon projection in Kir2.1-off mice.

of mitral/tufted cells (*Figure 5A–i*). The convergent pattern provides a structural basis to transmit odor information. Suppressing spontaneous activity and odor exposure both have led to divergent innervation of the TAAR4 expressing ORs. We thus examined whether the change in OSN axon projection patterns altered the specificity of connection between OSNs and the mitral/tufted cells. In one scenario, mitral/tufted cell dendrites are connected to OSNs in the same glomerulus regardless of the ORs they express (*Figure 5A–ii*). It is also possible that the connection between sensory neurons and postsynaptic mitral cells is specified by molecular identities independent of spatial locations. In this scenario, axons expressing the same OR may be matched to the same mitral cell even though they are distributed into many glomeruli (*Figure 5A–iii and iv*). The latter scenario would suggest the specific connection between OSNs and mitral/tufted cells could be maintained.

We noticed that in the Kir2.1-off mice axons expressing the same receptor were more concentrated in compartments within the glomeruli, with their terminal zones smaller than those observed in controls (*Figure 5B*). If the dendrites of mitral cells were to match the axons of a given OR identity, we would expect a similar compartmentalization of dendritic trees. Thus, we quantified the level of compartmentalization of both axons and dendrites. We estimated the fractions of a glomerulus that could be innervated by a single type of axons in the Kir2.1-off mice carrying the *Olfr160-IRES-tauGFP* (*M72-IRES-tauGFP*) allele. We used antibodies against GFP to label *Olfr160* (*M72*) expressing axons and stained against olfactory marker protein (OMP) or neural cell adhesion molecule (NCAM) to label the glomeruli (*Keller and Margolis, 1975*; *Terkelsen et al., 1989*). In control mice, glomeruli innervated by the M72 axons were filled with GFP expressing fibers and exhibited near complete signal overlap between GFP and OMP, or GFP and NCAM (*Figure 5B–i and ii* and *Figure 5—figure supplement 1A-i and ii*). In Kir2.1-off mice, many glomeruli receiving M72 fibers contained regions that were OMP+ (*Figure 5B–iii and iv*) or NCAM+ (*Figure 5—figure supplement 1Aiii and iv*) but GFP−, indicating that the glomeruli containing the M72 axons were innervated by axons expressing other receptors. Quantitative analysis revealed that glomeruli exhibited varied levels of innervation by the GFP-labeled axons for M72-expressing neurons (*Figure 5B–v* and *Figure 5—figure supplement 1B*) even though the glomerular size remained constant (*Figure 5B–vi*).

If the mitral cell dendrites were to match the axons, their projection pattern must be compartmentalized as well. To compare the level of compartmentalization, we analyzed glomerular occupancy by OSN axons and by mitral cell dendrites (*Figure 5C and D*). To distinguish the two patterns, we applied a series of spatial filters of increasing sizes to tile the glomeruli and quantified the number of filters containing fluorescent signal (*Figure 5C and D*). As filter size increased, glomerular occupancy increased and eventually reached 100% when the filter reached the size of the glomerulus. This rate of increase was fast for projections that distributed widely, but slow for those concentrated in small areas. This analysis showed that the axons from Kir2.1-off mice were compartmentalized. At filter size of 6 pixels, these axons occupied ~40% of the glomeruli compared to 100% in the control (*Figure 5C–iii and iv*). The dendrites of the mitral cells, in contrast, did not project to more than one glomerulus. Within the glomerulus, they projected broadly into all parts of the glomeruli (*Figure 5D*) in both wild-type and Kir2.1-off mice. They showed identical distribution and the same rates of increase with filter size (*Figure 5D-iii and iv*). The mismatch between the axon and dendritic projection patterns indicated that in the Kir2.1-off mice, OSNs expressing a particular OR

were connected with different mitral cells and vice versa. Individual mitral cells were likely to receive input from multiple receptor types. Because odor identities are encoded by the population response of mitral/tufted cells, a change in the connection between the OSNs and the mitral cells, therefore, would alter the identity of the odors intrinsically embedded by a genetic program.

## Transcriptomic changes in response to silencing and activation

The observation that silencing OSNs and persistent odor exposure altered axon projection raised the question as to what the mechanism was in driving these changes. In particular, it was puzzling that PEA exposure would increase the number of TAAR4 glomeruli as odor experience had been shown to facilitate the convergence of axons (*Zou et al., 2004*). Neural activity influences gene expression in the OSNs during development (*Connelly et al., 2013*; *Inoue et al., 2018*; *Nakashima et al., 2019*; *Serizawa et al., 2006*). Specific patterns of activity have been associated with the expression of axon guidance molecules that determine target specificity of the OSNs (*Nakashima et al., 2013*). We therefore examined the impact of neural activity on gene expression during the early postnatal period.

Analysis of our previous single-cell RNA-Seq (scRNA-Seq) experiment indicated that many genes associated with axon guidance were detected at low levels in individual cells, likely due to the low read depth associated with the scRNA-Seq platform (*Wu et al., 2018*). Therefore, we performed bulk RNA-Seq of olfactory epithelia from Kir2.1 and PEA-exposed mice at P5, before the closure of the critical period. We have identified through differential expression analysis (adjusted p<0.05) 563 genes either upregulated or downregulated by Kir2.1 expression (*Figure 6A*). Gene ontology (GO) term analysis indicated highly enriched gene sets involved in the ion channels, synaptic connection, axon growth, and intracellular vesicles (*Figure 6B and C*). Among the OR genes, 43 were differentially expressed, all of which showed reduced expression (*Figure 6—figure supplement 1A*). Several genes known to be dependent on neural activity exhibited changes consistent with previous studies. For example, the homotypic attractive molecule *Kirrel2* and two homotypic repulsive cues *Epha5* and *Epha7* were downregulated, whereas *Kirrel3* was upregulated (*Figure 6D*; *Cutforth et al., 2003*; *Serizawa et al., 2006*). Several OR and *Taar* genes, including *Taar4* and *Taar7*, were also downregulated (*Figure 6F and G*). The changes of these genes could underlie the dysregulation of axon projection in the Kir2.1 mice.

PEA-exposure from P0 to P5 induced changes in 262 genes (*Figure 6E*). GO term analysis did not reveal any class of genes that was specifically enriched (*Figure 6B*). This could reflect the fact the PEA exposure only affects a small subset of neurons expressing cognate receptors for PEA. A systematic change of gene expression in these neurons might be too small to be statistically significant. A targeted analysis revealed change of expression for eight ORs, four of which were upregulated in the PEA group (*Olfr550*, *Olfr888*, *Olfr889*, and *Olfr890*; *Figure 6—figure supplement 1B*), and four were downregulated (*Olfr143*, *Olfr239*, *Olfr1325*, and *Olfr335-ps*; *Figure 6—figure supplement 1B*). This was in contrast to the Kir2.1 experiment, which had 43 downregulated OR genes.

There were only eight differentially expressed genes shared by both Kir2.1 and PEA treated data sets (*Figure 6F*; *Figure 6—figure supplement 1C and D*). Four of them were downregulated in both Kir2.1 and PEA-exposed groups (*Clstn2*, *Cntn4*, *Olfr1325*, and *Olfr239*). Besides the olfactory receptors, two other genes were known to be involved in axon guidance. *Contactin*4 (*Cntn4*) is a cell-adhesion molecule that exhibits discrete levels of expression for different receptor types in the olfactory glomeruli (*Kaneko-Goto et al., 2008*). *Clstn2* is a member of the *Calsyntenin* family of genes that has been implicated in regulating axon guidance at choice point (*de Ramon Francàs et al., 2017*). Four genes showed changes in the opposite direction, downregulated in the Kir2.1 mice but upregulated in the PEA treated group (*Ccdc33*, *Atp8b3*, *Icmt*, and *Kifl7*). Although none of these genes have a known function in axon guidance, *Kif7* is a kinesin family member implicated in the development of zebrafish photoreceptors (*Lewis et al., 2017*). None of the *Taar* genes were significantly changed in expression at the statistical level, but there was a clear trend of downregulation of *Taar4*, *Taar7b*, and *Taar9* (*Figure 6G*). The *Taar5* gene, encoding a receptor for TMA, did not have an obvious change. The downregulation of *Taar4* by PEA exposure was consistent with a previous study (*Dewan et al., 2018*).

Taken together, the transcriptomic analysis indicated that Kir2.1 expression and PEA exposure had resulted in changes in several guidance molecules that reflected a change in neural activities and connectivity. These changes may explain the divergent projection of OSN axons.

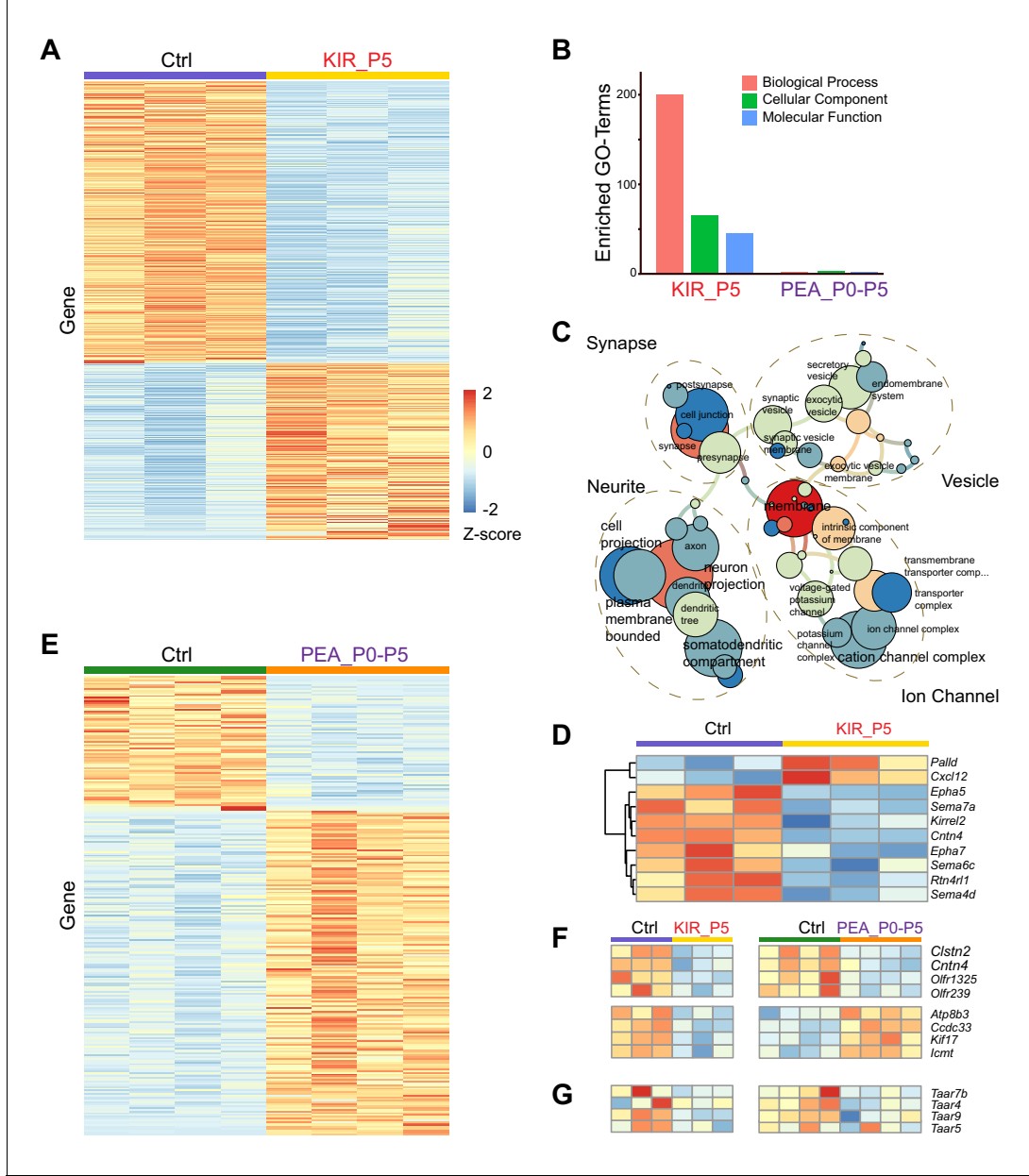

**Figure 6.** Transcriptomic changes modulated by neural activity. (**A**) Heatmap of differentially expressed genes in the olfactory epithelia between control and Kir2.1 mice at P5. Each row is a gene. Each column is from one mouse. Color indicates Z-score. (**B**) Number of enriched GO term of differentially expressed genes for Kir2.1 and 2-phenylethylamine (PEA) data. (**C**) GO term analysis of the Kir2.1 data. GO term enrichment is represented as a hierarchical map built from differentially expressed genes in the group. The size of the circle represents –log2 p-value derived from Fisher's test. Colors indicate the level of connectivity of a GO term. Only major GO terms are shown to simplify the graphs. (**D**) Heatmap showing differentially expressed genes associated with axon guidance in the Kir2.1 data. (**E**) Heatmap of genes differentially expressed between control and PEA-treated mice. (**F**) Heatmaps showing the expression of eight genes common to both Kir2.1 and PEA data. (**G**). Heatmap showing the expression of four *Taar* genes in the two data sets. Note that the expression of these four genes were not identified as statistically significant.

The online version of this article includes the following figure supplement(s) for figure 6:

**Figure supplement 1.** Differentially expressed genes.

Downregulation of the OR genes including *Taar4* may indirectly alter the expression of axon guidance molecules.

## Altered central representation of innately recognized odors

If the odor identities were altered, we hypothesized that the valence associated with these odors would be changed such that the brain nuclei that represented the innate valence of these odors would not be activated. Odor information is processed by multiple pathways in the brain. Whereas the anterior olfactory nucleus and the piriform cortex are involved in channeling odor information regardless of their valence, other brain areas were associated with odor valence. Positive valence is generally associated with the medial amygdaloid nucleus (postdorsal area; MePD and posteroventral area; MePV) (*Boehm et al., 2005*; *Kang et al., 2009*; *Yoon et al., 2005*). Negative valence is generally associated with the bed nucleus of stria terminalis (BST) (*Duvarci et al., 2009*; *Kim et al., 2013*) and the anterior hypothalamic area (AHA) (*Canteras et al., 2001*; *Gross and Canteras, 2012*). The posterolateral cortical amygdaloid area (PLCo) and the ventral medial hypothalamus (VMH) have been implicated in both positive and negative valence (*Brennan and Kendrick, 2006*; *Kollack-Walker and Newman, 1995*; *Lin et al., 2011*; *Root et al., 2014*; *Wang et al., 2015*). We therefore examined the activation of these brain regions by the innately recognized odors. We performed immunofluorescence staining using antibodies against phosphorylated ribosomal protein S6 (pS6) ribosomal protein after exposing the animals to a specific odor (*Knight et al., 2012*). 2-MBA activated the BST, AHA, and VMH over background when compared with no stimulus in the control mouse brains (*Figure 7A and C*). However, we did not observe above background activation in the Kir2.1-off mouse brain. The aversive odor PEA activated these brain regions in the Kir2.1-off animals to a much less extent than control (*Figure 7—figure supplement 1*). We also observed that the PLCo, MePD, MePV, and VMHvl were activated by female urine in the control mice but not the Kir2.1-off mice (*Figure 7B and D*). These observations suggested that the Kir2.1-off mice had an altered representation of the odors in brain regions associated with valence.

## Altered odor representation by early odor exposure

We next examined the activation of brain regions associated with innate odor-triggered behaviors in animals exposed to PEA during the critical period. We found that in mice that were exposed to PEA during the first 2 weeks after birth, PEA no longer activated BST, AHA, BLA, and MePV above background (*Figure 8A*). In contrast, in mice exposed to PEA after the critical period, these brain areas were activated by PEA (*Figure 8*). These findings were consistent with the change of innate aversion toward PEA.

# Discussion

## Plasticity in an innate sensory circuit

Naïve animals recognize ethologically relevant odors such that they can engage in social interactions, find food, and avoid predators without prior experiences. The discovery that the circuit for innate odor perception is malleable shows that circuits processing innate responses are not fully hardwired in the mammalian brain. They are subject to influence by both spontaneous and experiential activities. Importantly, this plasticity is only observed in the early postnatal period when the olfactory map is still developing and when projection patterns are susceptible to different perturbations but retain their capacity to recover when perturbations are removed (*Ma et al., 2014*; *Tsai and Barnea, 2014*; *Wu et al., 2018*). These findings show that as a general principle, neural activities regulate the development of sensory circuits in the mammalian nervous system.

Neural activities have a broad impact on the establishment of the neural circuits. The role played by neural activities in shaping the olfactory circuits appears different from that of other sensory systems. Visual input, for instance, is required for the segregation of ocular-specific input in the lateral geniculate and in the cortex (*Shatz and Stryker, 1978*; *Wiesel and Hubel, 1963*). Dark rearing delays the binocular segregation, as well as the closure of the critical period in the visual cortex. In contrast, spontaneous activities, not odor exposure per se, are required for the convergence of axons (*Yu et al., 2004*). Past studies have shown that pairing an odor with an aversive unconditioned stimulus can lead to enlarged glomeruli corresponding to the cognate receptor, enhanced transmitter release at the OSN termini, and heightened sensitivity to odor stimulation in the interneuron population (*Bhattarai et al., 2020*; *Dias and Ressler, 2014*; *Jones et al., 2008*; *Kass and McGann, 2017*). Early exposure of odorants can also lead to increased expression of OR genes and enlarged

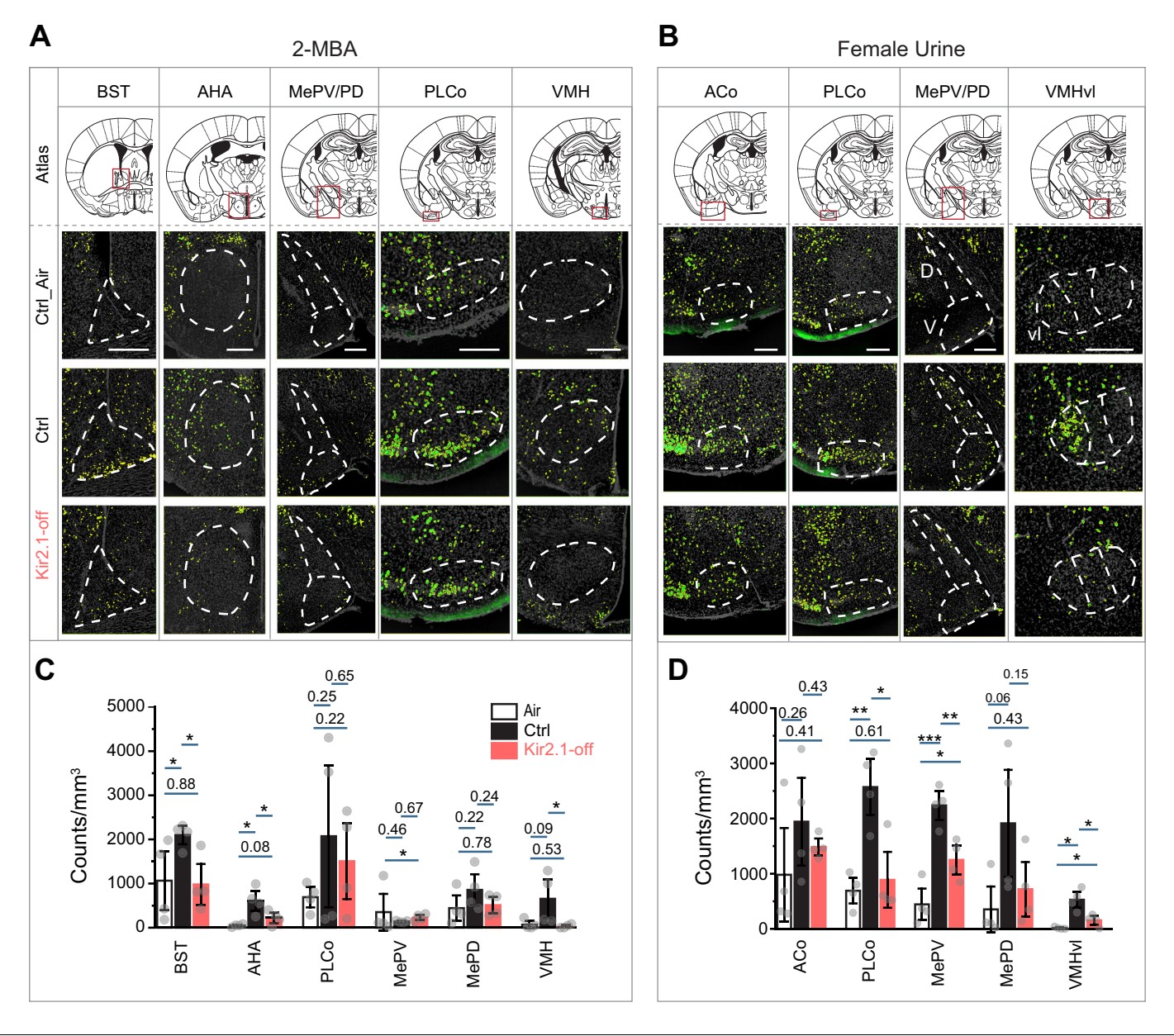

**Figure 7.** Suppression of spontaneous activities during development alters representation of innate odors in the adult brain. (**A and B**) Immunofluorescent staining of phospho-S6 (green) of brain sections from Ctrl and Kir2.1-off animals treated with 2-MBA Green channel is enhanced to make cells visible (**A**) and female urine (**B**). Cell nuclei are counterstained with DAPI (gray). Top panel shows the atlas maps (adapted from The Mouse Brain Stereotaxic Coordinates) (**Paxinos and Franklin, 2013**), with the boxes indicating the brain regions in the lower panels. Scale bar, 300 μm. (**C and D**) Bar plots show the density of activated cells in different brain areas in Ctrl (**C**) and Kir2.1-off mice (**D**) (data are shown in mean ± SEM, n = 4 hemispheres). Individual data points are shown as gray dots. One-way ANOVA performed pairwise comparison. *, p<0.05; **, p<0.01; ***, p<0.01. The online version of this article includes the following source data and figure supplement(s) for figure 7:

**Source data 1.** Quantification of cells activated by innate odors in various brain regions in control and Kir2.1-off mice.
**Figure supplement 1.** Altered representation of innate odors in the brain by suppression of spontaneous activity.
**Figure supplement 1—source data 1.** Quantification of cells activated by 2-phenylethylamine in various brain regions of KIr2.1-off mice.

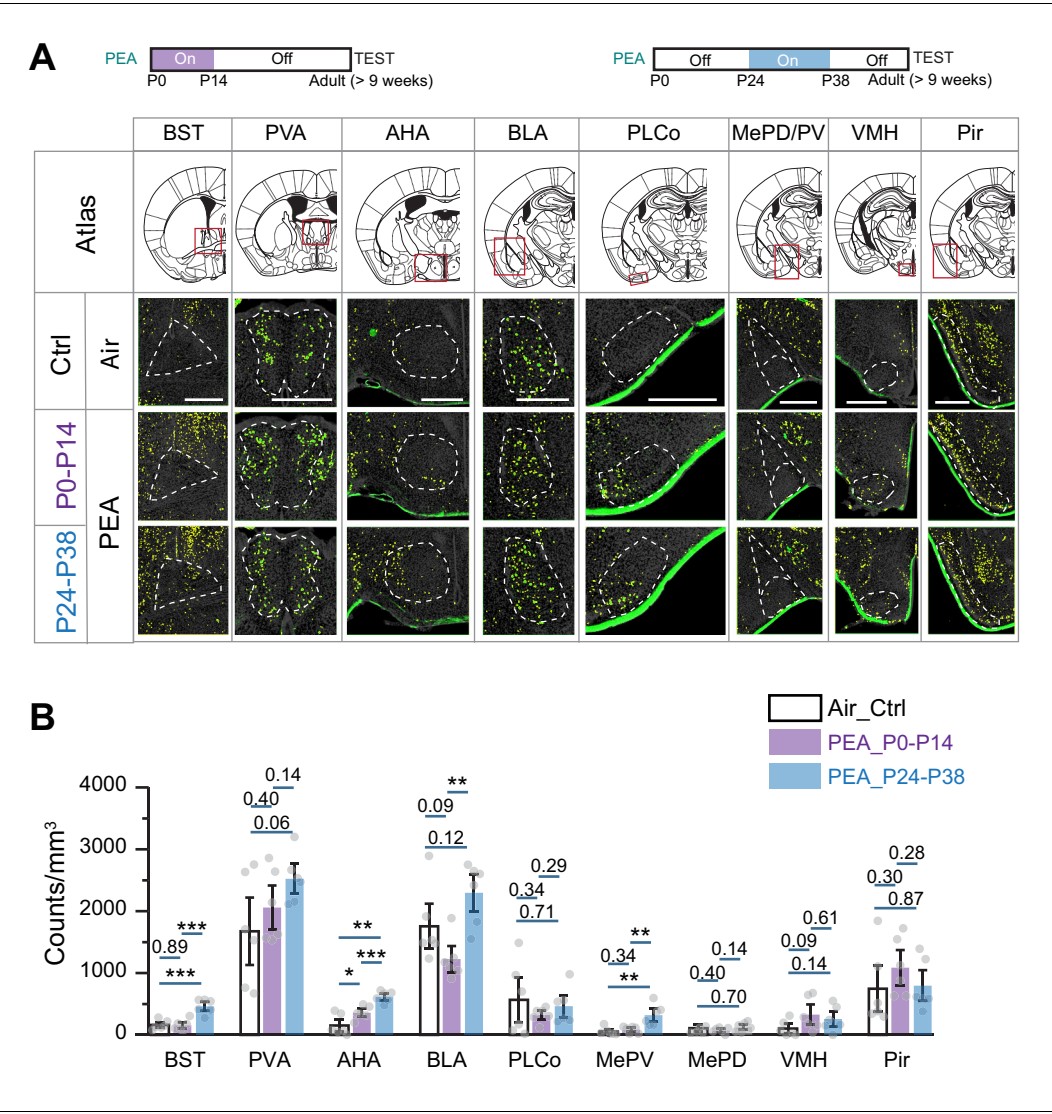

**Figure 8.** Early postnatal exposure to aversive odor alters representation of innate odors in the adult brain. (**A**) Immunofluorescent staining of phospho-S6 (green) of brain sections from 2-phenylethylamine (PEA)-treated mice. Cell nuclei are counterstained with DAPI (gray). Scale bar, 500 μm. (**B**) Bar plot shows the density of activated cells in different brain areas (data are shown in mean ± SEM, n = 6 hemispheres). One-way ANOVA was applied to pairwise comparison. *, p<0.05; **, p<0.01; ***, p<0.01.

The online version of this article includes the following source data for figure 8:

**Source data 1.** Quantification of cells activated by innate odors in various brain regions in mice with postnatal odor exposure.

olfactory glomeruli (*Cadiou et al., 2014*; *Ibarra-Soria et al., 2017*; *Todrank et al., 2011*). In this study, we further show that persistent stimulation by an innately aversive odor can lead to broadened axon projection and alter the innate valence of the odor.

The molecular mechanisms that specify connections between the OSNs and the mitral/tufted cells are not fully understood. A set of guidance molecules have been proposed to drive the sorting and convergence of axons expressing the same OR genes (*Kaneko-Goto et al., 2008*; *Nakashima et al., 2013*; *Takeuchi et al., 2010*; *Takeuchi and Sakano, 2014*). The expression of these molecules is activity dependent, relying on ligand-independent activation of the ORs, as well as specific firing patterns of OSNs (*Cadiou et al., 2014*; *Ibarra-Soria et al., 2017*; *Nakashima et al., 2019*; *Nakashima et al., 2013*; *Zhao et al., 2013*). In this study, a broad survey of transcriptomic changes in Kir2.1 mice revealed that suppression of the spontaneous activity of the neurons alters the

expression of specific guidance and cell adhesion molecules in the OSNs. The data is consistent with previous reports of expression level changes associated with neural activities (*Nakashima et al., 2019*; *Zhao et al., 2013*). Odor exposure primarily activates a small set of neurons expressing the cognate receptors. Activation of these neurons may change guidance molecules in these neurons, leading to altered axon projection. On the other hand, prolonged exposure may also lead to habituation and reduced activities. In our experiments, PEA exposure does not induce a broad change in gene expression associated with axon guidance as the Kir2.1 expression does. Nonetheless, it affects the expression of a small subset of genes that are also downregulated in the Kir2.1 mice. Although it is not clear how PEA exposure influences the expression of these genes at a similar level as Kir2.1, the changes are specific to only to these few genes. It will be interesting to investigate the mechanism by which the changes occur. Notably, *Contactin4* and *Clstn2* genes are downregulated by both silencing of the OSNs and PEA exposure. Although *Clstn2* has been implicated in axon guidance, its role in the olfactory system is not known. Our data indicate that it may serve as a potential regulator of olfactory axon guidance. Our transcriptome analysis provides support for a mechanism in which silencing neurons, or selectively activating some neurons, changes the expression of guidance molecules by overriding the activity patterns dictated by ligand-independent activities of ORs, thereby affecting projection patterns.

## Glomerular convergence, odor sensitivity, and discrimination

A previous study reported olfactory defects when examining mice with ectopic expression of Kir2.1 (*Lorenzon et al., 2015*). The observations were likely caused by the continuous expression of Kir2.1 channel in OSNs, which not only suppressed the spontaneous activities of the neurons but also reduced odor-evoked responses. In our study, by contrast, the Kir2.1-off mice were fed with DOX after the critical period to restore neural activities. These experiments were not complicated by the effect of suppressed neuronal responses.

In the Kir2.1-off mice, the olfactory map was imprinted from the early developmental stage. The axon projection patterns were permanently altered from single receptor type convergence into a pattern with individual glomerulus receiving innervation from multiple receptor type axons. These mice provided a unique opportunity to test the contribution of the single receptor type convergence on odor sensitivity, discrimination, and association. In wild-type mice, errors in the convergence of axons expressing the same OR into their perspective glomeruli are rarely observed. In both wild-type and in mice with genetic manipulations to alter axon projection patterns, ectopic glomeruli can be pruned during development (*Ma et al., 2014*; *Tsai and Barnea, 2014*; *Zou et al., 2004*). A robust mechanism that maintains the convergent projection pattern has likely been selected during evolution to serve important functions, including odor recognition and discrimination. It is therefore surprising to find that Kir2.1-off mice do not have major defects in odor detection, discrimination, and association. The single receptor type convergence does not appear to be required for these functions. In light of its role in innate odor perception (see below), we reason that one function of the convergent olfactory map is to provide consistency in odor perception among individuals. Alternatively, the convergent map may offer an advantage in fine odor discrimination that is not captured by our current study (*Gronowitz et al., 2021*).

## Plasticity in innate odor perception

Adaptive changes in innate odor perception are likely shaped by evolution to accommodate specific living environments. Amines are commonly found in urine, but different species excrete different types. PEA is enriched in carnivore urine and is avoided by rodents (*Ferrero et al., 2011*). Rats also avoid TMA but mice do not (*Li et al., 2013*). The divergent response in the rodent species has been proposed to be associated with the divergent biosynthesis pathway. Mouse urine contains >1000 × the concentration of TMA than rat. Whereas evolution may have led to divergent neuronal connections that contribute to the differential response, it is also possible that the exposure to TMA after birth eliminates aversion in mice. The change in valence is bi-directional as exposure to TMA at a higher concentration eliminates TMA attraction in the exposed mice.

Recent studies have demonstrated adaptive responses to semiochemicals in the vomeronasal organ and in the main olfactory system through sexually dimorphic and activity-dependent alteration of responsive cell types (*Vihani et al., 2020*; *Xu et al., 2016*). In the main olfactory system, this

change has been attributed to changes in the expression of ORs. Our study reveals that the altered response to PEA is associated with divergence of the axon projection of the TAAR4 expressing OSNs. Although this is a decrease in TAAR4 expression following PEA exposure, the change is not significant enough to change detection and discrimination of PEA in the adults.

Our results show that perturbing the convergent pattern through either transient silencing or activation of the OSNs is associated with altered innate behavioral responses to both attractive and aversive odors. Because Kir2.1 mediated neuronal silencing affects not only the OSNs, but also the vomeronasal neurons. This raises the possibility that the loss of attraction to urine could be attributed to the loss of pheromone sensing as a teaching signal. While this remains a possibility, it does not explain the loss of innate responses to food or predator odors.

The divergent projection of OSNs does not affect the single glomerular projection of the primary dendrites of the mitral/tufted cells. This result reaffirms the previous observations that OSN axons and mitral cell dendrites develop independently and are highly stable (*Cao et al., 2007*; *Inoue et al., 2018*; *Ma et al., 2014*; *Mizrahi and Katz, 2003*). Furthermore, it extends the previous studies by showing that it is unlikely to be a strict molecular identity code that matches OSN axons with mitral cell dendrites as the two show different compartmentalization patterns in the glomeruli. This observation is consistent with a recent study showing that mitral cell dendrites mostly project to the spatially proximal glomeruli (*Nishizumi et al., 2019*). Taken together, these results indicate that glomerular convergence is critical for innate coding of odor identity. It is likely that specific sets of glomeruli and the mitral/tufted cells innervating these glomeruli carry information about the identities of innately recognized odors. This stereotyped connection provides a substrate for the alteration of innate preference by altering the projection patterns of OSN axons.

Notably, although OSN axons project to multiple glomeruli in the Kir2.1-off as well as in odor-exposed mice, they maintain their general dorsal–ventral and anterior–posterior position in the bulb. For odors activating receptor neurons that innervate the dorsal area of the glomeruli, this broad spatial target is not sufficient to assign negative valence to the odors and generate aversive responses, even though the dorsal bulb is shown to be required to convey negative valence information for odorants (*Kobayakawa et al., 2007*).

## Glomerular convergence and innate valence

How is valence information encoded for innately recognized odors? Our data show that altered innate responses are associated with changes in the convergence pattern of axons of OSNs in an odorant specific manner. Although the association may not be causal, the evidence strongly suggests that the divergent projection is a common mechanism that alters altered innate odor recognition. Broad changes in the convergent pattern by suppression of spontaneous activity impact all odors we examined. On the other hand, divergent of TAAR4 axons is related to neutralized perception of PEA, but not other odors. It is likely that the activation of a specific set of glomeruli activates the mitral/tufted cells that are connected with brain centers that control appetitive or aversive behaviors (*Figure 9A*). Perturbation of the projection pattern of OSN axons disrupts this association. Our experiments demonstrate that the mitral cells maintain the connection with individual glomeruli without discriminating the different types of axons. One possibility is that the mitral cells maintain specific connections with downstream behavioral centers in such a way that divergent innervation of glomeruli activates additional mitral/tufted cell populations, leading to the activation of brain centers that drive opposing behaviors (*Figure 9B*). This antagonism may lead to the loss of innate preference associated with specific odors.

An alternative, but not mutually exclusive scenario is that the change in OSN projection patterns leads to the recoding of odor identity (*Figure 9C*). Odor identities are represented by population responses of the mitral/tufted cells. The patterns of activation are not strictly determined by the direct relay from activated glomeruli but by the result of interactions among cells within the neuronal network. Innately recognized odors are represented by specific sets of neurons whose connections to the valence centers are strongly biased toward one appetitive or aversive pathway. This association, while genetically specified, is flexible and can also be modified. Developmental perturbations could alter the way odor identities are represented by activating a different set of neurons in the olfactory pathway. An altered representation would eliminate the biased activation of brain centers driving appetitive or aversive behaviors (*Figure 9C*). In some case, it may also create new valence association. This model is consistent with emerging evidence that suggests innate preference is

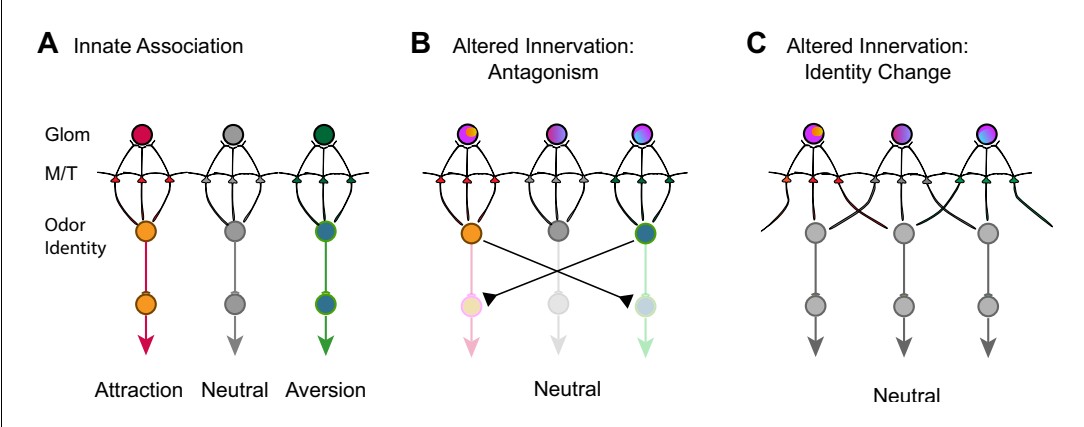

**Figure 9.** Models of encoding innate valence of odors. (**A**) Wild-type connections model. The activation of specific glomeruli leads to the activation of specific sets of mitral/tufted cells that connect to behavioral centers. Color indicates different associated valence. For simplicity, only single glomeruli for each valence is depicted. Odor identities are abstracted as single nodes. (**B**) Altered axon innervation model. Activation pattern with altered axon projection leads to the activations of brain centers that drive opposing behaviors and abolished innate responses normally associated with the same odor. Rainbow colors of the glomeruli indicate mixed axon innervation. (**C**) Alternative altered axon innervation model. The change in activity pattern of the mitral/tufted cells is not the result of direct relay from the glomerular activation but the result of intercell interactions within the olfactory bulb. This change results in the recoding of odor identities (gray nodes) by the mitral cells, leading to the appetitive or aversive responses to be abolished.

context dependent and has strong interactions among olfactory pathways (*Qiu et al., 2021*; *Saraiva et al., 2016*). It is also consistent with evidence that ethologically relevant odors are encoded similarly as ordinary odors in cortical structures thought to encode innate preference (*Iurilli and Datta, 2017*).

It is possible, even likely, that neuronal connections in the brain areas associated with odor identity and valence are influenced by spontaneous neuronal activity and odor experience. For example, the innervation of additional glomeruli may create association of an odor with brain areas to assign new valence. These changes may occur independently of the changes in the olfactory bulb. Our current study does not address these possibilities but provides a basis for further investigations.

# Materials and methods

## Key resources table

| Reagent type (species) or resource | Designation | Source or reference | Identifiers | Additional information |
|---|---|---|---|---|
| Strain, strain background (*M. musculus*) | C57BL/6J | The Jackson Laboratory | RRID: IMSR_JAX:000664 | |
| Strain, strain background (*M. musculus*) | CD1 | The Jackson Laboratory | RRID: IMSR_JAX: 003814 | |
| Strain, strain background (*M. musculus*) | *Omp-IRES-tTA* | The Jackson Laboratory | RRID: IMSR_JAX: 017754 | |
| Strain, strain background (*M. musculus*) | *tetO- Kcnj2-IRES-tauLacZ* | The Jackson Laboratory | RRID: IMSR_JAX: 009136 | Also known as: tetO- *Kir2.1*-IRES-tauLacZ |
| Strain, strain background (*M. musculus*) | *Olfr160-IRES-tauGFP* | The Jackson Laboratory | RRID: IMSR_JAX:006678 | Also known as: M72-IRES-tauGFP |
| Other | Amyl acetate | Sigma-Aldrich | CAS# 628-63-7 | Abbr: AA |
| Other | Hexanal | Sigma-Aldrich | CAS# 66-25-1; | Abbr: HXH |
| Other | 2-Pentanone | Sigma-Aldrich | CAS# 107-87-9 | Abbr: PTO |
| Other | Valeraldehyde | Sigma-Aldrich | CAS# 110-62-3 | Abbr: VAH |
| Other | Butyl acetate | Sigma-Aldrich | CAS# 123-86-4 | Abbr: BAE |

*Continued on next page*

*Continued*

| Reagent type (species) or resource | Designation | Source or reference | Identifiers | Additional information |
|---|---|---|---|---|
| Other | 2-Heptanone | Sigma-Aldrich | CAS# 110-43-0 | Abbr: HPO |
| Other | Methyl butyrate | Sigma-Aldrich | CAS# 623-42-7 | Abbr: MBE |
| Other | Methyl propionate | Sigma-Aldrich | CAS# 554-12-1 | Abbr: MPE |
| Other | R(−)-Carvone | Sigma-Aldrich | CAS# 6485-40-1 | Abbr: (−) Car |
| Other | S(+)-Carvone | Sigma-Aldrich | CAS# 2244-16-8 | Abbr: (+) Car |
| Other | Methyl caproate | Sigma-Aldrich | CAS# 106-70-7 | Abbr: MCE |
| Other | Heptanal | Sigma-Aldrich | CAS# 591-78-6 | Abbr: HPH |
| Other | Methyl valerate | Sigma-Aldrich | CAS# 624-24-8 | Abbr: MVE |
| Other | 2-Phenylethylamine | Sigma-Aldrich | CAS# 64-04-0 | Abbr: PEA |
| Other | Eugenol | Sigma-Aldrich | CAS# 97-53-0 | Abbr: EUG |
| Other | 2-Methylbutyric acid | Sigma-Aldrich | CAS# 116-53-0 | Abbr: 2-MBA |
| Other | Isoamylamine | Sigma-Aldrich | CAS# 107-85-7 | Abbr: IAMM |
| Other | Trimethylamine | Sigma-Aldrich | CAS# 1184-78-7 | Abbr: TMA |
| Other | Female mouse urine | Fresh collected | Strain: c57bl/6 | Abbr: FU |
| Other | Male mouse urine | Fresh collected | Strain: c57bl/6 | Abbr: MU |
| Other | Coyote urine | Harmon scents | CCHCY4 | Abbr: CU |
| Other | Peanut butter | Jif Extra Crunchy | | Abbr: PB |
| Other | Maple flavor | Frontier natural products co-op | #23081 | Abbr: Maple |
| Other | Lemon flavor | Frontier natural products co-op | #23071 | Abbr: Lemon |
| Antibodies | Donkey anti-Rabbit 488 | Thermo Fisher Scientific | R37118; RRID:AB_2556546 | (1:1000) |
| Antibodies | Rabbit anti-NCAM | Millipore Sigma | ABN2181-100UG | (1:400) |
| Antibodies | Goat anti_OMP | Wako | 544–10001; RRID:AB_2315007 | (1:400) |
| Antibodies | Rat anti-NCAM | Millipore | AB5032 | (1:400) |
| Antibodies | Chicken anti-GFP | Abcam | ab13970-100; RRID:AB_300798 | (1:1000) |
| Antibodies | Rabbit anti MOR28 | Gilad Barnea Lab | RRID:AB_2636804 | (1:500) |
| Antibodies | Guinea pig anti-TAAR4 | Gilad Barnea Lab | N/A | (1:500) |
| Antibodies | Rabbit anti-Phospho-S6 (Ser235/236) | Cell Signaling | 4854; RRID:AB_390782 | (1:1000) |
| Antibodies | Goat anti-Guinea Pig Alexa Fluor 488 | Thermo Fisher Scientific | A-11073; RRID:AB_2534117 | (1:1000) |
| Antibodies | Goat anti-Guinea Pig Alexa Fluor 568 | Thermo Fisher Scientific | A-11075; RRID:AB_2534119 | (1:1000) |
| Antibodies | DAPI | Thermo Fisher Scientific | D1306; RRID:AB_2629482 | (1:1000) |
| Antibodies | TOTO-3 | Thermo Fisher Scientific | T3604 | (1:1000) |
| Antibodies | Dextran, Biotin, 3000 MW, Lysine Fixable | Thermo Fisher Scientific | D7135 | |
| Antibodies | Streptavidin, Alexa Fluor 568 conjugate | Thermo Fisher Scientific | S11226; RRID:AB_2315774 | (1:1000) |
| Other | Doxycycline diet | Envigo | TD.120046 | |
| Other | C and B Metabond Quick Adhesive Cement System | Parkell | Cat# UN1247 | |

*Continued on next page*

*Continued*

| Reagent type (species) or resource | Designation | Source or reference | Identifiers | Additional information |
|---|---|---|---|---|
| Other | 3 p peptide | United Peptide | biotin-QIAKRRRLpSpSLR ApSTSKSESSQK | 25 nM |
| Other | Isoflurane | Patterson Veterinary | 07-893-1389; CAS# 26675-46-7; | 2–5% |
| Other | Xylazine | Patterson Veterinary | 07-893-2121 | 10 mg/kg |
| Other | Ketamine | Vedco | VINV-KETA-0VED | 100 mg/kg |
| Software and algorithms | OriginPro | Origin Lab | RRID:SCR_014212 | https://www.originlab.com/Origin |
| Software and algorithms | MATLAB | Mathworks | RRID:SCR_001622 | https://www.mathworks.com/ |
| Software and algorithms | Custom MATLAB scripts | Yu Lab | N/A | |
| Software and algorithms | ImageJ (Fiji) software | NIH | RRID:SCR_002285 | |
| Software and algorithms | R 4.0.3 | R Project for Statistical Computing | RRID:SCR_001905 | https://www.r-project.org/ |
| Software and algorithms | Salmon | https://combine-lab.github.io/salmon/ | | |
| Software and algorithms | ENSEMBL | https://www.ensembl.org/index.html | RRID:SCR_002344 | |
| Software and algorithms | tximport | https://www.bioconductor.org/packages/release/bioc/html/tximport.html | RRID:SCR_016752 | |
| Software and algorithms | DESeq2 | https://bioconductor.org/packages/release/bioc/html/DESeq2.html | RRID:SCR_015687 | |
| Software and algorithms | GOSeq | https://bioconductor.org/packages/release/bioc/html/goseq.html | RRID:SCR_017052 | |
| Software and algorithms | biomaRt | https://bioconductor.org/packages/release/bioc/html/biomaRt.html | RRID:SCR_019214 | |
| Software and algorithms | Gephi | https://gephi.org/ | SCR_004293 | |
| Software and algorithms | pheatmap | https://rdrr.io/cran/pheatmap/ | RRID:SCR_016418 | |
| Software and algorithms | rgexf | https://gvegayon.github.io/rgexf | | |

## Animals

The *tetO-Kcnj2-IRES-tauLacZ* (tetO- *Kir2.1*-IRES-tauLacZ), *Omp-IRES-tTA*, *Olfr160-IRES-tauGFP (M72-IRES-tauGFP*; Jackson laboratory, stock number 009136, 017754, 004946, and 006678, respectively) were described previously (*Barnea et al., 2004*; *Bozza et al., 2004*; *Feinstein and Mombaerts, 2004*; *Yu et al., 2004*). For early odor exposure experiments, CD1 (Jackson laboratory, stock number: 003814) animals were used. All animals were maintained in Lab Animal Services Facility of Stowers Institute at 12:12 light cycle and provided with food and water ad libitum except animals for two-choice and Go/No Go experiments. All behavior experiments were carried out during the dark cycle of the animals under red or infrared light illumination. Experimental protocols were approved by the Institutional Animal Care and Use Committee at Stowers Institute (protocol 2019–102) and in compliance with the NIH Guide for Care and Use of Animals.

Compound heterozygotes carrying the *Omp-IRES-tTA* and the *tetO-Kcnj2-IRES-taulacZ* alleles were weaned at P21 and fed with diet (Envigo) containing 20 mg/kg DOX for more than 6 weeks prior to any experiments unless otherwise indicated (Kir2.1-off). Single-allele littermates of the Kir2.1-off mice and wild-type C57BL/6J (Jackson laboratory, stock number: 000664) strain mice (referred to as B6) were subject to the same treatment and served as controls.

## Odor delivery with olfactometer

Odor delivery was controlled by an automated olfactometer with custom written software developed in the National Instrument Labview programming environment as described previously (*Ma et al., 2012*; *Qiu et al., 2014*). Odors, with the exception of TMA, were freshly prepared in mineral oil at desired concentration. The amines were purchased as free base form. PEA at 1:100 dilution in mineral oil has a pH value of 10.4. TMA was diluted in water, pH 11.2 at 85 mM. Urine and natural odors were used at original concentration. Odorants are listed in the Key Resource Table.

## Innate odor preference test

Innate odor preference tests were same as previously described (*Qiu et al., 2014*). 2–4 months old mice were used for experiments. Each experimental group contained 6–14 animals. Unless otherwise stated, all animals were naïve to the testing odors and exposed to the same odor once. Each animal was tested with a total of two odors in two separate experiments with at least 1 week between tests. After being habituated to the testing environment for half an hour, the animals were put into a 20 × 20 cm chamber for behavioral experiments. The chamber had a nose cone on one of the side wall 5 cm above the base plate. Odors were delivered through the nose cone by the olfactometer. A vacuum tube connected on the opposite wall of the nose cone provided an air flow to remove residual odors after odor delivery. Pure odorants were diluted into mineral oils at 1:1000 (v/v) in most cases. 10 ml/min air flow carried the saturated odor out from the odor vial and was further diluted into a 90 ml/min carrier air to make the final dilution to $10^{-4}$ (v/v). Delivery time, concentration, and sequence of odor delivery were controlled by the olfactometer. Investigation of odor source was registered by infrared beam breaking events and recorded by the same software that controlled the olfactometer.

Odor was delivered for 5 min in each trial with a 5-min interval. After four trials of air (over mineral oil vial) presentation, a testing odor was presented four times. In a typical test, mice habituating to the test chambers over the multiple sessions of background air led to decreased $T_{Air}$. The presentation of an odor elicited investigations and measured as $T_{Odor}$. If the odor is attractive to the animal, an increased in $T_{Odor1}$ is expected to be higher than that for a neutral odor as both novelty seeking and attraction drive the investigations. If the odor is aversive to the animal, a smaller increase or even decrease $T_{Odor1}$ is expected as the mixed result of novelty seeking (risk assessment) and avoidance, while the $T_{Odor2}$ is expected as the aversion only because the novelty is habituated quickly while the avoidance persists longer. We define the preference index:

$$\text{Preference index} = \frac{T_{Odor1} + T_{Odor2} - 2 * T_{Air4}}{T_{Ave.Air}} \times 100$$

## Cross habituation test

Cross habituation test was performed similar to innate odor preference tests with modification of odor presentation period. In each trial an odor was delivered for 1 min followed by 4 min of carrier air. Mice were first habituated with eight trials of air followed by testing odors. Investigations to the odor port during the control air presentation ($T_{Air}$) or odor presentation ($T_{Odor}$) were recorded. After the mice had been habituated to the first odor (five trials), a second odor was presented three to five times. In a typical test, mice habituating to the test chambers over the multiple sessions of background air led to decreased $T_{Air}$. The presentation of an odor elicited an increased $T_{Odor}$. Repeated presentation of the same odor led to habituation, which was reversed by the test odor if it was perceived as novel. No increased investigation was expected if the test odor was perceived similar to the habituating odor. Normalized NPI for individual trial was calculated by dividing $T_{Air}$ or $T_{Odor}$ by the average duration of odor port investigation during the background air $T_{Ave.Air}$ ($T_{Ave.Air} = \sum T_{Air}/N$). Distance between two odors in behavioral test was calculated as the difference in normalized exploration duration:

$$\Delta NPI = \frac{T_{odor2.1} + T_{odor1.5}}{T_{Ave.Air}} \times 100$$

where $T_{Odor2.1}$ is the exploration duration for the first session of Odor 2 (the testing odor) and $T_{Odor1.5}$ is the exploration time of the fifth (last) session of Odor 1 (the habituating odor). As such, $\Delta NPI$ is the difference between the investigation time of a novel odor and that of the habituated

odor, expressed as the percentage over the average exploration time during air presentation. Pairwise *t*-test was performed for statistical analysis.

## Dishabituation test for threshold determination

The trial sequence used to measure detection thresholds is the same as previous described (*Qiu et al., 2014*). After the eight initial air presentations, an odor was presented at the lowest concentration ($10^{-8}$ v/v). Two air presentation and an odor presentation interleave in the test with the odors being presented at increasing concentration. The p-values of normalized ΔNPI between air and the odor presentation were calculated.

ΔNPIs at different odor concentrations (*x*) were fitted with Nonlinear Least Squares method using Weibull psychometric function in MATLAB:

$$SR = A - (A - g) \times e^{-\left(\frac{kx}{t}\right)^b}$$

where

$$k = -\left[ln\left(\frac{A-a}{A-g}\right)\right]^{\frac{1}{b}}$$

*A* is the maximum performance, *b* is the steepness of the function, and *a* is the threshold concentration. The parameter *g* is the false alarm rate which is set at 0%.

Because PEA elicits aversive response from the animals, we only tested PEA for detection threshold at low concentrations. After the eight initial air presentations, PEA was presented at $10^{-13}$ mol/l, followed by interleaving two air presentations and one PEA presentation at $10^{-12}$ mol/l and $10^{-11}$ mol/l. Because we could not normalize to high concentration responses, ΔNPI values were used to determine detection threshold.

## Two alternative choice test and Go/No Go test

Animals were maintained on a water restriction regimen with 1.5 ml of drinking water per day for at least 1 week before training. Food was available ad libitum. A decrease in body weight of 10–15% from the original was normally observed. All the training and testing were performed during the dark/light cycle 24 hr after the last water feeding.

For two alternative choice (two-choice) test, water restricted animals were introduced into a 20 × 20 cm behavioral box that contains three nosecones on one side of the wall. Each nosecone contains a pair of infrared emitter and receiver and the nose poke events were registered as the breaking of IR beam. The central port delivers an odor puff whereas the two side ports delivered water. Delivery was triggered when the animal poked in. Odor delivery, nose poke event registration, and the water reward were controlled by the olfactometer. In initial training, animals were presented with a single odor that was associated with a fixed water port. Once the animals learned to go to the water port upon odor delivery in the odor port, they were subject to two-choice training. Two odors were delivered in a pseudo random sequence and water reward was contingent upon the animal correctly associating the odor with the appropriate water port. If the animal chose the correct port, 0.05 ml water was released as reward for the right choice. Otherwise, no water was given. The animals were considered to have learned the behavior paradigm if the success rate reached 80%. Success rate (SR) was calculated as:

$$SR = \frac{P_{[A/A]} + P_{[B/B]}}{P_{total}} \times 100$$

where $P_{[A/A]}$, $P_{[B/B]}$, and $P_{total}$ are the number of pokes into A water port upon delivery of odor A, the number of pokes into B water port upon odor B, and the total number of nose pokes into the odor port, respectively. Upon reaching criteria, the concentrations of both A and B were gradually decreased for the two-choice test.

Go/No Go assay was applied for odor threshold detection as a second method as described previously (*Qiu et al., 2014*). Basically, water deprived animals were trained to associate an CS+ odor (AA) with water reward by poking into the nose cone and licking the waterspout when the CS+ odor

delivered. A CS− presentation (air) was coupled with a mild electric shock (30 v, 0.7 s), when the mouse licked the waterspout. Success rate (SR) was calculated as:

$$SR = \frac{P_{CS+} + NP_{CS-}}{P_{CS+} + NP_{CS+} + P_{CS-} + NP_{CS-}}$$

where $P_{CS+}$ and $NP_{CS+}$ are the number of licking and non-licking events for the CS+ odor, $P_{CS-}$ and $NP_{CS-}$ are number of the licking and non-licking events for the CS− odor respectively. After the animals were considered to have learned the behavior paradigm when success rate reached 90%, we measured SR of licking events at different concentrations of AA. The measured performance values also were fitted by the Weibull psychometric function with the parameter $g$ was at the chance level (50%).

## Dendritic knob recording on M72-GFP OSNs

Cell-attached loose patch clamp recordings were done on intact sheets of dissected olfactory epithelium (OE). During the recording, the epithelium was perfused continuously with oxygenated ACSF and Ringer's solution in a custom-made recording chamber. The dendritic knobs of the M72-GFP neurons were identified and visualized using Zeiss LSM510 with excitation at 488 nm. Recordings were made using glass pipettes with tip size of less than 1 μm and resistance of 10–25 MΩ. The recording electrodes were filled with Ringer's solution. The action potentials were acquired using Multiclamp 700A (Molecular Devices) and were digitized at 10 kHz by Digidata 1440A (Molecular Devices). The data was recorded and analyzed using pClamp10 software (Molecular Devices).

## EOG recording

Half-brain preparations were freshly made by bisecting the head through the midline to expose the OE. A glass pipette (1 MΩ resistance) filled with 1× PBS was used in the EOG recording. AA was prepared at 1:100 dilution and delivered at 50 ml/min flow rate in 50 ml/min moisturized carrier air. Total flow rate was maintained at 100 ml/min during the experiment. Odor was delivered for 0.5 s to the epithelium followed by a 20 s interval. Average response from four trials was used in quantification. Signal was amplified with Axon CyberAmp320 (Molecular Devices) and digitized with Axon Digidata 1320 (Molecular Devices). The pClamp10 software (Molecular Devices) was used for data acquisition and processing. Each half brain was recorded for no more than 15 min to ensure the viability of the OSNs.

## Immunofluorescent staining and dye labeling

The mice carrying the *Olfr160-IRES-GFP* allele were crossed into the *Omp-IRES-tTA / tetO-Kcnj2-IRES-taulacZ* compound heterozygotic background and treated with DOX to obtain control or Kir2.1-off mice. Immunofluorescent staining was carried out using 14–16 μm olfactory bulb cryosections or 50 μm olfactory bulb sections with vibratome (Leica VT 1000 s) prepared from animals perfused with 4% paraformaldehyde (PFA). Sections were washed in 1× PBS, permeabilized in 1× PBS containing 0.1% TritonX-100 (PBST), and blocked in 1% skim milk dissolved in PBST. Rabbit or rat anti-NCAM (1:400), chicken anti-GFP (1:400), goat anti-OMP (1:400), rabbit anti-MOR28 (1:500), rabbit anti-Phospho-S6 (1:1000), and guinea pig anti-TAAR4 (1:500) antibodies in PBST were used to stain the section overnight at 4°C or room temperature. Alexa Fluor conjugated secondary antibodies (Donkey anti-Rabbit 488, Goat anti-Guinea Pig Alexa Fluor 488, and Goat anti-Guinea Pig Alexa Fluor 568) were diluted to 1:1000 in PBST for staining overnight. TOTO-3 or DAPI was used for nuclear staining.

Mitral cell dendritic tuft labeling was performed by iontophoresis method (Stimulus isolator A360, World Precision Instrument. 2.5 μA, 2 Hz, 5–10 min) of biotinylated dextran amines (BDA, 3000 MW, Life Technology) into the dorsal and median of mitral cell layers in olfactory bulb. Animals were perfused with 1× PBS with 4% PFA 12 hr after labeling. The olfactory bulb was dissected and post-fixed with 4% PFA at 4°C overnight, and then sectioned at 50 μm thickness with vibratome (Leica VT 1000 s). The sections were then stained with 2 μg/ml streptavidin Alexa Fluor 568 conjugate and DAPI.

All images were taken on a Zeiss LSM510, LSM700, or Zeiss 510 LIVE confocal system. For co-localization analysis, an image stack was scanned using multi-track method and a maximal projection of the stack was obtained. Co-localization coefficient was calculated as the percentage of GFP

positive pixels over the NCAM positive pixels within a glomerulus. For mitral cell labeling, Z-stack confocal images were obtained for dendrite tracing. For interneuron staining, cells in the glomeruli layer with positive staining were counted using ImageJ. Glomeruli numbers were calculated by DAPI staining.

Glomerulus occupancy were calculated using a custom-written script in ImageJ. Images were Z projected using maximum projection method in ImageJ and a hard threshold was applied to obtain binary images. After the glomerulus is circumscribed, a series of spatial filters with increasing pixel sizes (1–20 pixels) were applied to tile the glomeruli. Each filter containing a positive signal is considered to be occupied. Glomerular occupancy was quantified by computing the percentage of filters with a signal in the circumscribed glomerulus.

### Phosphorylated ribosomal protein S6 mapping of odor-evoked activity

For pS6 staining, animals were single housed and habituated in home cages for 7 days with a glass vial covered with a plastic cover, which was punched with seven holes for odor evaporation and avoiding physical contact to chemicals. For habituation, a small piece of cotton nestlet soaked with 500 µl mineral oil was put inside the vial. Vials were changed every day at 1 hr after light cycle. On day 8, new glass vial with cotton nestlet soaked with 500 µl PEA, 2-MBA at $1:10^3$ dilution in mineral oil, or freshly collected male or female urine was added in the home cages. One hour after odor stimulation, mice were sacrificed and intracardiac perfused with 4% PFA. The mouse brains were dissected and then post-fixed with 4% PFA overnight at 4°C.

The phospho-S6 immunochemistry histology was performed based on the published protocol (*Knight et al., 2012*) with some modifications. The entire brain was cut into 50 µm thick serial sections using a Leica vibratome (VT1000S). Rabbit anti-phospho-S6 antibody (1:1000 dilution) and the 3 p peptide (25 nM, synthesized by United Peptide and has the sequence biotin-QIAKRRRLpSpSL-RApSTSKSESSQK, where pS is phosphoserine) in 1× PBST were used at 4°C overnight. Tiled images were acquired using Olympus VS120 Virtual Slide Microscope, or PE Ultraview spinning disk confocal microscope (PerkinElmer) which were stitched together using the Volocity software (PerkinElmer). Different brain nuclei were identified based on the brain atlas (The Mouse Brain Stereotaxic Coordinates, third/fourth edition) (*Franklin and Paxinos, 2008*; *Paxinos and Franklin, 2013*). The pS6 immuno-positive neurons were counted using ImageJ. For quantification the two sides of the brain were treated independently and the following numbers of sections were used: anterior hypothalamic area, anterior part (AHA), four sections at −0.85 to −1.05 mm from Bregma; medial amygdaloid nucleus (postdorsal area; MePD and posteroventral area; MePV) and ventromedial hypothalamic nucleus (ventrolateral area; VMHvl), five sections between −1.35 and −1.60 mm from Bregma; hypothalamic nucleus (VMH) in the 2-MBA case, four sections between −1.80 and −2.00 mm from Bregma; bed nucleus of the stria terminalis (BST), three sections between 0.10 and 0.25 mm from Bregma; anterior cortical amygdaloid area (ACo), four sections at −1.00 to −1.20 mm from Bregma; posterolateral cortical amygdaloid area (PLCo), five sections at −1.35 to −1.60 mm from Bregma; paraventricular thalamic nucleus, anterior part (PVA), four sections at −0.25 to −0.45 mm from Bregma; piriform cortex (Pir), five sections at −1.35 to −1.60 mm from Bregma; basolateral amygdaloid nucleus, anterior part (BLA), five sections at −1.10 to −1.35 mm from Bregma.

### Odor exposure

For early odor exposure experiments, odorants are either soaked in a cotton nestlet (1 ml 1:100 PEA in mineral oil, or 85 mM TMA in water) or in solid form (1 g PB) placed in a glass vial inside the home cage. A single hole punched in the lid allows odor to diffuse. The odor vial was changed daily with fresh prepared odors. For the P0–14, the home cage contains both mother and pups. After the odor stimulation period, the animals were raised regularly without the odor vials till 9 weeks for behavior and histology experiments.

### Measuring odor concentration in home cage

Odor concentrations in home cage were measured by miniPID (200B, Aurora Scientific Inc). A glass vial containing 1 ml 1:100 v/v PEA was placed in the center of a clean cage. A baseline was collected from another clean cage without odor vial. Then the inlet of the miniPID was placed at various

distances from the odor vial opening. Signals from the miniPID were digitized by MiniDigi-1B (Molecular Devices) at 1 kHz and low pass filtered (10 Hz) in pCLAMP10.3 (Molecular Devices).

## Bulk RNA sequencing

Kir2.1 pups and their littermates, and CD-1 pups treated from birth and the control pups were sacrificed at postnatal day 5. Whole olfactory epithelia were removed from individual samples and deposited directly into TRIzol solution (Thermo Fisher Scientific). Total RNA was extracted using Direct-zol RNA miniprep plus (Zymo Research) according to the manufacturer's instructions. Sequencing libraries for the Kir2.1 experiment were generated with Illumina's mRNA-Seq Sample Prep Kit (Cat#RS-930–1001), using standard poly-A tail selection protocols, and were sequenced as 40 bp single-end unstranded reads on Illumina's C4D679 platform. Sequencing libraries for PEA-treated samples were generated using Illumina's TruSeq Stranded Total RNA Kit (Cat#RS-122-2301and Cat#RS-122–2302), which utilizes RiboZero Gold rRNA Removal. The libraries were subsequently sequenced as 150 bp single-end stranded reads on Illumina's NextSeq 500 platform. FASTQ files were generated using bcl2fastq v2.18. FastQC v0.11.7 was implemented to generate reports for each sample to ensure sequencing quality.

## Selective-alignment and transcript quantification

With Salmon v1.3.0 (*Patro et al., 2017*) we performed a selective alignment of the raw FASTQ reads to the *Mus musculus* transcriptome (GRCm38, ENSEMBL Release 100) (*Yates et al., 2020*). To minimize the potential for reads of genomic origin mapping spuriously to the transcriptome we included the whole reference genome as an alignment decoy for the creation of the Salmon index. We concatenated the whole genome and transcriptome FASTA reference files into a single file, 'gentrome.fa', and input this into the command, 'salmon index –`kmerLen` 31 –`transcripts gentrome.fa` –`decoys decoys.txt` –`index salmon_index`', where the k-mer length was 31, 'decoys.txt' was a one column list containing the identifiers for each chromosome or scaffold in the reference genome, and 'salmon_index' was the directory name for the output index. Estimates of transcript-level expression were generated for each sample FASTQ by calling 'salmon quant –`index salmon_index` –`libType SR` –`unmatedReads sample.fq` –`seqBias` –`gcBias`'. Here 'SR' indicated a reverse-stranded single-end read library, and the options 'seqBias' and 'gcBias' employed algorithms to correct for sequence-specific and GC biases, respectively. The output transcript-level count-estimates were subsequently grouped and summed on their ENSEMBL parent-gene identifiers for downstream gene-level analyses.

## Differential expression

Salmon count estimates were imported into R v4.0.3 with tximport v1.18.0 (*Soneson et al., 2016*). The PEA and Kir2.1 data sets were analyzed separately for differential gene expression with DESeq2 v1.30.0 (*Love et al., 2014*). The 'DESeq' function takes as input the raw gene counts, a data table of the sample IDs and independent variables, and the experimental design formula; it then normalizes the raw gene counts to account for variation in read-depth between samples, calculates gene-wise dispersion estimates, and finally implements a Wald test for the significance of each coefficient in the negative binomial generalized linear model. Tests for differential gene expression were performed with an assigned significance threshold of FDR $\leq$ 0.05. Estimates of $\log_2$ fold-change were shrunk using the *apeglm* method (*Zhu et al., 2019*). Heatmaps show z-scores for the normalized gene counts and were created with the R package pheatmap v1.0.12 (*Kolde, 2019*).

## GO analysis

Independent GO analyses were performed on the PEA and Kir2.1 data with the R package GOseq v1.40.0 (*Young et al., 2010*) using biomaRt v2.46.0 (*Durinck et al., 2009*) GO annotations. First, we calculated a probability weighting function (PWF) to estimate gene-length bias with the 'nullp' function given the inputs of a named binary vector with all significantly differentially expressed genes (DEGs) assigned one and all other tested genes assigned 0, and a named numeric vector of mean effective gene-lengths. p-values for GO category enrichment were calculated with the 'goseq' function, using the default 'Wallenius' method on the PWF. p-values were corrected using the Benjamini and Hochberg procedure and a significance threshold was set at FDR $\leq$ 0.05. Gephi v0.9.2

(*Bastian et al., 2009*) was used to visualize hierarchical networks of significantly enriched GO terms. The R package rgexf v0.16.0 (*Vega Yon et al., 2015*) was used to convert 'goseq' output into Gephi compatible files.

## Quantification and statistical analysis

All the statistics except for RNA-seq data are conducted in MATLAB or OriginPro. Data were expressed as means ± SEMs in figures and text. Group differences were analyzed using one-way ANNOVA with Turkey test unless otherwise specified. Significance was defined as: * indicates $p<0.05$, ** indicates $p<0.01$, *** indicates $p<0.001$.

# Acknowledgements

We thank A Moran and members of the Lab Animal Services, R Egidy, A Scott, K Zueckert-Gaudenz, M Peterson, Dr. A Perera, and members of the Molecular Biology Facility at the Stowers Institute for technical assistance. We are grateful to Dr. P Mombaerts for providing the *Olfr160-IRES-tauGFP* mice, and Dr. G Barnea for providing antibodies against TAAR4 and MOR28. We thank Dr. H Li for assistance in statistics. We also thank the members of the Yu laboratory for their valuable inputs.

# Additional information

## Funding

| Funder | Grant reference number | Author |
|---|---|---|
| National Institutes of Health | R01DC008003 | C Ron Yu |
| National Institutes of Health | R01DC014701 | C Ron Yu |
| National Institutes of Health | R01DC016696 | C Ron Yu |
| Stowers Institute for Medical Research | 1021 | C Ron Yu |

The funders had no role in study design, data collection and interpretation, or the decision to submit the work for publication.

## Author contributions

Qiang Qiu, Conceptualization, Resources, Data curation, Formal analysis, Validation, Investigation, Visualization, Methodology, Writing - review and editing; Yunming Wu, Conceptualization, Resources, Data curation, Software, Formal analysis, Validation, Investigation, Methodology, Writing - review and editing; Limei Ma, Conceptualization, Resources, Supervision, Investigation, Methodology, Project administration, Writing - review and editing; Wenjing Xu, Conceptualization, Resources, Supervision, Funding acquisition, Validation, Investigation, Methodology, Writing - original draft, Project administration, Writing - review and editing; Max Hills Jr, Data curation, Investigation, Visualization, Methodology, Writing - review and editing; Vivekanandan Ramalingam, Investigation, Methodology; C Ron Yu, Conceptualization, Resources, Data curation, Formal analysis, Supervision, Funding acquisition, Validation, Visualization, Methodology, Writing - original draft, Project administration, Writing - review and editing

## Author ORCIDs

Qiang Qiu https://orcid.org/0000-0002-7280-1005
C Ron Yu https://orcid.org/0000-0003-1555-8683

## Ethics

Animal experimentation: Experimental protocols were approved by the Institutional Animal Care and Use Committee at Stowers Institute (protocol 2019-102) and in compliance with the NIH Guide for Care and Use of Animals.

Decision letter and Author response
Decision letter https://doi.org/10.7554/eLife.60546.sa1
Author response https://doi.org/10.7554/eLife.60546.sa2

## Additional files

### Supplementary files

• Transparent reporting form

### Data availability

Sequencing data have been deposited in GEO under accession codes GSE166457. All other data generated or analysed during this study will be available at http://www.stowers.org/research/publications/libpb-1613.

The following datasets were generated:

| Author(s) | Year | Dataset title | Dataset URL | Database and Identifier |
|---|---|---|---|---|
| Qiu Q, Wu Y, Ma L, Xu W, Hills MH, Ramalingam V, Yu CR | 2021 | Acquisition of Innate Odor Preference Depends on Spontaneous and Experiential Activities During Critical Period | https://www.ncbi.nlm.nih.gov/geo/query/acc.cgi?acc=GSE166457 | NCBI Gene Expression Omnibus, GSE166457 |
| Qiu Q, Wu Y, Ma L, Xu W, Hills MH, Ramalingam V, Yu CR | 2021 | Acquisition of innate odor preference depends on spontaneous and experiential activities during critical period | http://www.stowers.org/research/publications/libpb-1613 | Stowers Institute Original Data Repository, 1613 |

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
