## [Decision Letter]

**Acceptance summary:**

This manuscript puts forward the interesting idea that olfactory sensory neuron activity during a critical period modulates innate behavioral responses to certain odors, such as attractive food odors/pheromones and aversive predator odors.

**Decision letter after peer review:**

Thank you for submitting your article "Acquisition of Innate Odor Preference Depends on Spontaneous and Experiential Activities During Critical Period" for consideration by *eLife*. Your article has been reviewed by 3 peer reviewers, one of whom is a member of our Board of Reviewing Editors, and the evaluation has been overseen by Piali Sengupta as the Senior Editor. The following individual involved in review of your submission has agreed to reveal their identity: Julian P Meeks (Reviewer #3).

The reviewers have discussed the reviews with one another and the Reviewing Editor has drafted this decision to help you prepare a revised submission.

All reviewers thought the basic observations of Figures 1/2 to be interesting, that innate aversion responses require the proper level of activity during an early olfactory critical period. However, there was substantial concern that the proposed mechanism involving altered glomerular targeting was not well supported, with data presented being largely correlational. It was noted that the odor exposure experiments were particularly surprising and distinct for responses seen to other odors previously.

In preparing your revision, reviewers after discussion highlighted the need for additional experiments to support mechanistic conclusions that activity manipulations alter innate odor responses through aberrant glomerular targeting. Reviewers also called for a similar analysis of attractive odor responses. Finally, several textual revisions are requested, as detailed in the full reviews below, which we hope are helpful to you.

Reviewer #1:

This manuscript puts forward the interesting idea that olfactory sensory neuron activity during a critical period modulates innate behavioral responses to certain odors, such as attractive food odors/pheromones and aversive predator odors. This is a novel concept, and an interesting topic, as the neural basis underlying innate odor responses in the mouse remains poorly investigated. The manuscript in its current form presents a lot of data, albeit mostly correlational. It seems that either increasing or decreasing sensory neuron activity can perturb innate responses- these findings suggest that a delicate activity balance is important, yet underlying mechanisms are not clear. In general data related to behavior and sensory neuron projections (Figures 1-4) are more compelling than cFos data and arguments for mitral cell connectivity (Figures 5-8). Overall, there are some novel findings that will be interesting to the field, but mechanistic uncertainty remains.

1. I am surprised that critical period odor exposure broadens connectivity patterns to multiple glomeruli. This seems counterintuitive, as activity-dependent refinement is thought to promote target selectivity. How could this be occurring mechanistically?

2. On a related note, mice are presumably supplied with the TAAR5 agonist (trimethylamine) in their rearing cage from parental odors. This should cause normal spreading of TAAR5 glomeruli in wt mice- from prior publications though (for example Johnson, Barnea 2012), there do not seem to be more TAAR5 glomeruli than TAAR4 glomeruli, but maybe this should be re-examined with existing antibodies. How general is this phenomenon of odor-evoked connectivity broadening- does it extend to odorant receptors too?

3. There is substantial evidence that attraction to volatile urinary cues and to food odors is learned rather than innate. Perhaps loss of volatile pheromone attraction is due to loss of attraction to the UCS darcin – if darcin is no longer attractive, then it cannot serve as a positive-valence teaching signal to entrain volatile odor attraction. In this model, Kir2.1 manipulations would really be having their effect on darcin-activated vomeronasal circuits, which are not investigated or at least discussed.

4. As a side note, the model put forward could very well explain how 'innate responses' vary across species. For example, mice do not avoid the urinary odor/TAAR5 agonist trimethylamine while rats do. Since trimethylamine is abundant in mouse urine, perhaps early exposure helps override what would otherwise be a developmentally induced aversion. Likewise, carnivore odors such as PEA likely evoke different reactions in mice than in the carnivore species that produce them, and similar mechanisms could be at play to help ensure species-appropriate responses. It may be interesting to discuss further.

Reviewer #2:

Qiu et al. examined the consequences of suppressing neuronal activity (via OMP-tTA x tetO-Kir2.1 mice) during the development of the olfactory epithelium on innate odor preferences. They found that suppressing neuronal activity abolished such preferences. Additionally, exposure to an aversive odorant during development abolished innate aversive in adults. Both manipulations led to altered glomerular targeting without affecting responses and behaviors neutral odorants. These manipulations also affected the number of cells in downstream brain regions that were activated by innate odors. Combined, this manuscript contains interesting findings about the plasticity of innate odor responses, extends the author's prior work on critical periods for glomerular targeting, and provides new data on the functional consequences of manipulating olfactory system activity during early developing. I find the observation that early manipulations seem to specifically abolish innately valenced responses to odors to be incredibly important- it seems like an experiment someone should have done long ago, and this will be an significant contribution to the literature. This experiment opens up a lot of future work and will potentially cause real rethinking in the field. On the other hand, the main weaknesses here are mechanistic – there isn't a clear story that emerges either about how valence is established, or about how the specific manipulations abolish innate odor valence.

1. The link between the two experiments used throughout this manuscript (suppressing activity in OMP-tTA x tetO-Kir2.1 mice and early exposure to PEA) is conceptually unclear. In the former, the authors suppressed neuronal activity early in development, whereas in the latter activity is presumably increased via the daily exposure. However, both manipulations resulted in the same behavioral phenotype (the lack of avoidance in adults) as well as similar deficits in glomerular targeting. Although the authors provide models of how the changes in OB wiring could lead to the observed results, further discussion of how these changes emerge (e.g. alterations in axon guidance molecules) and how these opposing manipulations could lead to the same anatomical and behavioral phenotypes is warranted.

2. How does innate attraction work given these results? Meaning – the old Sullivan paper suggests that early odor exposure assigns appetitive valence to even "aversive" odors. But the PEA experiment doesn't explain to me how positive valence is assigned, it just tells me how to break valence assignment. What is going on there with respect to these findings? In an ideal world, to strengthen these findings the authors should repeat the PEA experiments with appetitive odorants and show that they observe the same effects; this will also show that the results from the TAAR system generalize to other ORs.

3. The results of the early PEA exposure experiments are intriguing, but further work is necessary to interpret the conclusions from these experiments. First, the authors use a concentration of 1:100 when exposure to μM concentrations (10,000 times less than what was used here) has been shown to suppress Taar4 transcript levels (Dewan 2018). Furthermore, the behavioral results in Figure 2 currently phenocopy the loss of activity in the Kir2.1 animals and the effects in TAAR4-KO animals (Dewan 2013), so it would be good to verify that the manipulations in the odor exposure experiments are indeed elevating activity levels. Therefore, the authors should monitor in more detail the physiological consequences of their early exposure paradigm (e.g. using the methods described in Figure 1B-C) and specifically assess that the high concentrations of odorants used are not reducing the OR levels in TAAR4-expressing neurons or altering the function of these neurons.

4. As the authors noted, it is surprising that surprising that the Kir2.1-off mice do not have major defects in odor detection, discrimination and association. Does this apply to the innate odorants (which can generally be reassigned meaning through reward): is detection and or discrimination impaired for any the innate odorants? Likewise, is the detection of PEA impaired after the early exposure paradigm?

5. It would be helpful if the authors could discuss their findings in the context of prior work studying the consequences of odor exposures on olfactory system plasticity and OB glomeruli. As an example, other odor exposure paradigms have resulted in increases in glomerular volume, changes in OSN numbers, or changes in OSN neurotransmitter release, among other effects (see e.g. Kass 2013, Bhattarai 2020, Jones 2008, Todrank 2010, Cadiou 2014, Ibarra-Soria 2017, Xu 2016). The results described in this manuscript differ from some of these; therefore, a more comprehensive discussion of prior work could be useful to contextualize and contrast the findings in this manuscript. There is also prior work on OSN-M/C targeting (e.g. Inoue 2018) and M/C dendritic stability (e.g. Mizrahi and Katz 2003).

6. The introduction seeks to define differences between innate and learned behaviors, and map this onto predicted differences between circuits; many of the statements that end up being made are both overly didactic and highly debatable. To take just a single example on page 4 – "circuits that underlie innate responses are thought to be insensitive to sensory experiences." Obviously both worm chemoattraction and fly courtship are innate, hardwired and deeply plastic in response to sensory experience. The first three paragraphs have this quality (e.g., innate behaviors are not the same as fixed action patterns, it is not at all required that circuits that support innate behaviors be insulated from other types of information, what about retinal waves, etc). My advice is pretty simple – just say that we think of behaviors as innate because in adult animals we observe these behaviors and animals execute them without apparent prior experience or training. What this paper does is to go back and ask whether these innate behaviors actually depend upon early experience or neural activity; if so, that has implications for possible circuit mechanisms. Simple will clarify here.

Reviewer #3:

The work by Qiu et al. describes experiments using transgenic mice in which spiking is selectively downregulated in OMP-expressing olfactory sensory neurons (OSNs) during the pre-weaning period. The authors find that innate attractive and aversive behavioral responses are lacking in these mice in adulthood. They also show that tonic presentation of an innately aversive odorant during the early postnatal period causes the loss of aversion to that odorant later in life. Both sets of behavioral results appear to correlate with changes in glomerular convergence and the activation of downstream brain regions associated with olfactory valence.

The strength of this work is that it supports a critical role for early olfactory experience in future assignment of odor valence. Though many of the mechanisms underlying this phenomenon remain unknown/unexplored, this represents a compelling set of data and an important topic in the field. There are some notable limitations to these experiments that, if addressed in text or with inclusion of additional data or analysis, would improve confidence in the conclusions.

1) In the Discussion (pp 23-24), the authors state: "Our results demonstrate the importance of single glomerulus axon convergence in providing a substrate for innate odor recognition." This is implying a causative role for the glomerular convergence in the observed changes in valence. This is too strong. These events are correlated, but it does not necessarily follow that the lack of glomerular convergence causes loss of innate valence (while supporting odorant discrimination).

2) The results in Figure 4 are used to support the claim that DOX+ mice are still able to perform odor discrimination in adulthood. However, these experiments are all on neutral odors even though the central conclusion of the manuscript relates to odors with innate valence. It would improve this core conclusion if the manuscript included data on olfactory discrimination using odors with innate valence. For example, it would be useful to know whether DOX+ adult mice can discriminate PEA (which no longer causes innate aversion for these mice) from neutral or attractive odors.

3) The experiments present an interesting dichotomy: selectively silencing all OMP+ OSNs or constitutively activating OSNs that sense PEA both result in a loss of innate valence in adulthood. This suggests that the pattern of early postnatal activity in OSNs influences the animal's capacity to express an innate behavioral response to odorants later in life. This is exciting, but begs the question: does chronic postnatal exposure to an innately attractive odor (e.g. peanut oil) result in the same effect? If so, this might provide support for a unified mechanism of valence disruption (e.g. Figure 8). If not, this would still be interesting and would add depth to the study. At a minimum this should be discussed.

4) The analysis of spiking following DOX treatment in OMP-tTA x tetO-Kir2 mice seems to indicate a subtle, but potentially important change in the pattern of DOX+ OSN spiking towards burst-like firing. The ISI distribution appears to be skewed towards lower ISIs, but the analysis related to Figure 1B (a Student's t-test) indicates a lack of statistical significance. A Student's t-test does not seem to be an appropriate test for these non-normally distributed data. Also, there remain fairly meager data related to the transition from near-silence of OSNs in the DOX- case to the DOX+ post-weaning conditions. If this period involves transitions from near-silence to widespread bursting in all OMP+ neurons prior to a mildly bursty adult steady-state, this would seem to be important.

---

## [Author Response]

All reviewers thought the basic observations of Figures 1/2 to be interesting, that innate aversion responses require the proper level of activity during an early olfactory critical period. However, there was substantial concern that the proposed mechanism involving altered glomerular targeting was not well supported, with data presented being largely correlational. It was noted that the odor exposure experiments were particularly surprising and distinct for responses seen to other odors previously.In preparing your revision, reviewers after discussion highlighted the need for additional experiments to support mechanistic conclusions that activity manipulations alter innate odor responses through aberrant glomerular targeting. Reviewers also called for a similar analysis of attractive odor responses. Finally, several textual revisions are requested, as detailed in the full reviews below, which we hope are helpful to you.

It is known that suppressing neural activities alter axon projection patterns by dysregulating cell adhesion molecules expressed by the OSNs. In our study, persistent odor exposure during early development alters the projection of neurons expressing the cognate receptor for the odor, but not others. This is likely due to increased neural activity in these cells, which can also alter their axon targeting. To understand whether the two manipulations share a common mechanism, we performed transcriptome analysis of the olfactory epithelium at postnatal day 5 for control, Kir2.1, and PEA exposed pups. As expected, suppressing neural activity by Kir2.1 expression caused a broad change in the transcriptome associated with neurite growth, synapse formation, ion channels, and intracellular vesicles. Several known molecules involved in olfactory axon guidance were differentially regulated. This was consistent with the observed change in axon projection for all OSN types examined. On the other hand, PEA treatment also caused transcriptome change but there was no GO term standing out. We did observe, however, that a small subset of differentially expressed genes were shared by the two datasets. It included two molecules involved in axons guidance, Calsyntenin 2 and Contactin 4. These new results provide an explanation of how opposite manipulations of neural activity may lead to similar changes in the axon projection pattern. We also have attempted to perform antibody staining to address whether the changes are specific for the TAAR-expressing glomeruli in the PEA-exposed animals. However, the problem with antibodies have prevented us to get this result. We have expanded discussion of potential mechanism by which neural activities drive aberrant glomerular targeting.

Per recommendation, we also conducted experiment using attractive odors, including the urinary compound TMA, and food odor (peanut butter). In both cases, early postnatal exposure diminished innate preference in an odor-specific manner. Thus, the observation is a general one applying to odors of either valence.

Reviewer #1:This manuscript puts forward the interesting idea that olfactory sensory neuron activity during a critical period modulates innate behavioral responses to certain odors, such as attractive food odors/pheromones and aversive predator odors. This is a novel concept, and an interesting topic, as the neural basis underlying innate odor responses in the mouse remains poorly investigated. The manuscript in its current form presents a lot of data, albeit mostly correlational. It seems that either increasing or decreasing sensory neuron activity can perturb innate responses- these findings suggest that a delicate activity balance is important, yet underlying mechanisms are not clear. In general data related to behavior and sensory neuron projections (Figures 1-4) are more compelling than cFos data and arguments for mitral cell connectivity (Figures 5-8). Overall, there are some novel findings that will be interesting to the field, but mechanistic uncertainty remains.1. I am surprised that critical period odor exposure broadens connectivity patterns to multiple glomeruli. This seems counterintuitive, as activity-dependent refinement is thought to promote target selectivity. How could this be occurring mechanistically?

The reviewer correctly points out that sensory experience has been shown to refine target selectivity. Besides examples in the visual and auditory systems, naris occlusion has been shown to delay the pruning of ectopic projections in the olfactory system. Our data with Kir2.1-off mice is consistent with these studies. On the other hand, exposure to a single odorant does not increase sensory-triggered activity across all OSNs, but selectively activates the OSNs expressing the cognate receptors. Only these neurons project to additional glomeruli. This is analogous to an expanded representation of stimuli that are over-represented in the environment, such as the expanded representation of fingers in violin or piano players. We have included the discussion to clarify how OSN axon projection is influenced by neural activities.

The mechanism that drives the divergent projection is multifold and is an active area of research in our lab. For this study, we have now included transcriptome data, which shows that both silencing and odor exposure have led to dysregulated expression of axon guidance molecules. This provides an explanation as to how the divergent projection takes place. The new data is presented in the new Figure 6.

2. On a related note, mice are presumably supplied with the TAAR5 agonist (trimethylamine) in their rearing cage from parental odors. This should cause normal spreading of TAAR5 glomeruli in wt mice- from prior publications though (for example Johnson, Barnea 2012), there do not seem to be more TAAR5 glomeruli than TAAR4 glomeruli, but maybe this should be re-examined with existing antibodies. How general is this phenomenon of odor-evoked connectivity broadening- does it extend to odorant receptors too?

We appreciate this point. According to Li et al., 2013, trimethylamine (TMA) is at a relatively low concentration in female urine (~0.25mM) but high in male urine (~5mM). As the pups mostly grow up with the dams, they are likely exposed to the lower concentration. We have performed experiments to expose the pups to TMA (presented at 85 mM in vial, corresponding to ~5mM in cage). The early exposure abolishes TMA preference in adults. Whether there is an increase in TAAR5 glomeruli in response to TMA is an interesting question. The number of glomeruli innervated by a single type of OSNs varies between the ORs and between animals (Zapiec and Mombaerts, PNAS 2015). In the case of TAAR5, because the animals develop in an environment constantly exposing to TMA, it is not known what the number of glomeruli “should” be in an odor-free environment. We have attempted to examine whether there is a difference in the number of glomeruli between TMA exposed and control animals by antibody staining of the TAAR5 glomeruli. Unfortunately, multiple attempts including using new batches of antibodies and using different protocols were to no avail. It is likely that the antibodies have expired.

Although we cannot verify a change of the TAAR5 glomeruli in response to TMA exposure, we have verified this observation with other receptors. In a separate study, we have found that in mice exposed to acetophenone, a ligand for the M71 and M72 receptors, OSNs expressing these two receptors also project to multiple glomeruli. Thus, this observation is common to all the receptors we have examined. As this new data is an integral part of another manuscript, in which we show that the Wnt receptor Fzd1 is required to remodel the ectopic projections, we won’t present this result in the revised manuscript.

3. There is substantial evidence that attraction to volatile urinary cues and to food odors is learned rather than innate. Perhaps loss of volatile pheromone attraction is due to loss of attraction to the UCS darcin – if darcin is no longer attractive, then it cannot serve as a positive-valence teaching signal to entrain volatile odor attraction. In this model, Kir2.1 manipulations would really be having their effect on darcin-activated vomeronasal circuits, which are not investigated or at least discussed.

The reviewer raises an interesting point. It is possible that silencing the neurons during early development disrupts pheromone perception, which in turn disrupts the association of urinary odor with positive valence. In this case, the loss of attractiveness of urine could be secondary to perturbation of pheromone sensing. We now have discussed this possibility.

4. As a side note, the model put forward could very well explain how 'innate responses' vary across species. For example, mice do not avoid the urinary odor/TAAR5 agonist trimethylamine while rats do. Since trimethylamine is abundant in mouse urine, perhaps early exposure helps override what would otherwise be a developmentally induced aversion. Likewise, carnivore odors such as PEA likely evoke different reactions in mice than in the carnivore species that produce them, and similar mechanisms could be at play to help ensure species-appropriate responses. It may be interesting to discuss further.

We appreciate this suggestion. The species difference may arise from evolutionary advantages for the detection of species-specific cues. We have included the discussion on this point.

Reviewer #2:Qiu et al. examined the consequences of suppressing neuronal activity (via OMP-tTA x tetO-Kir2.1 mice) during the development of the olfactory epithelium on innate odor preferences. They found that suppressing neuronal activity abolished such preferences. Additionally, exposure to an aversive odorant during development abolished innate aversive in adults. Both manipulations led to altered glomerular targeting without affecting responses and behaviors neutral odorants. These manipulations also affected the number of cells in downstream brain regions that were activated by innate odors. Combined, this manuscript contains interesting findings about the plasticity of innate odor responses, extends the author's prior work on critical periods for glomerular targeting, and provides new data on the functional consequences of manipulating olfactory system activity during early developing. I find the observation that early manipulations seem to specifically abolish innately valenced responses to odors to be incredibly important – it seems like an experiment someone should have done long ago, and this will be an significant contribution to the literature. This experiment opens up a lot of future work and will potentially cause real rethinking in the field. On the other hand, the main weaknesses here are mechanistic – there isn't a clear story that emerges either about how valence is established, or about how the specific manipulations abolish innate odor valence.

We appreciate the encouraging assessment. It has been thought that innate valence is genetically hardwired and is communicated through labeled lines. In a related study (in press at Current Biology), we show that valence information is unlikely transmitted through labeled lines, but through population codes at various processing stages. These results are in line with the study by Illui and Datta showing that the innately recognized odors are not overly represented in the posterolateral cortical amygdala, the “valence center”. Nevertheless, the innate nature of odor valence association suggests that some genetic component must be involved. The current study shows that its establishment requires spontaneous neural activity and can be altered by sensory experience. These results suggest that the connectivity establishing innate valence is more complex than labeled lines. We are actively engaged in pursuing this question.

1. The link between the two experiments used throughout this manuscript (suppressing activity in OMP-tTA x tetO-Kir2.1 mice and early exposure to PEA) is conceptually unclear. In the former, the authors suppressed neuronal activity early in development, whereas in the latter activity is presumably increased via the daily exposure. However, both manipulations resulted in the same behavioral phenotype (the lack of avoidance in adults) as well as similar deficits in glomerular targeting. Although the authors provide models of how the changes in OB wiring could lead to the observed results, further discussion of how these changes emerge (e.g. alterations in axon guidance molecules) and how these opposing manipulations could lead to the same anatomical and behavioral phenotypes is warranted.

To understand whether the two manipulations share a common mechanism, we performed transcriptome analysis of the olfactory epithelium at postnatal day 5 for control, Kir2.1, and PEA exposed pups. As expected, suppressing neural activity by Kir2.1 expression caused a broad change in the transcriptome associated with neurite growth, synapse formation, ion channels, and intracellular vesicles. Several known molecules involved in olfactory axon guidance were differentially regulated. This was consistent with the observed change in axon projection for all OSN types examined. On the other hand, PEA treatment also caused transcriptome change but there was no GO term standing out. We did observe, however, that a small subset of differentially expressed genes shared by the two datasets. It included two molecules involved in axon guidance, Calsyntenin 2 and Contactin 4. These new results provide an explanation of how opposite manipulations of neural activity may lead to similar changes in axon projection pattern. The data are presented in the new Figure 6. Although we wished to address this question at a finer detail, by examining the changes specifically at the TAAR4 and TAAR5 glomeruli, we were unsuccessful because the antibodies stopped working. These results also raise new questions as to how these transcriptomic changes are induced by neural activity, which we intend to study.

2. How does innate attraction work given these results? Meaning – the old Sullivan paper suggests that early odor exposure assigns appetitive valence to even "aversive" odors. But the PEA experiment doesn't explain to me how positive valence is assigned, it just tells me how to break valence assignment. What is going on there with respect to these findings? In an ideal world, to strengthen these findings the authors should repeat the PEA experiments with appetitive odorants and show that they observe the same effects; this will also show that the results from the TAAR system generalize to other ORs.

We appreciate this question and the excellent suggestion. PEA exposure may have changed valence (measured as approach behavior) by one of two mechanisms. One is that the innate association between odor identity and valence is perturbed. This would be similar to the results observed in the neuronal silencing experiments. Another possibility is that through exposure, the aversive odor also acquires positive valence, as suggested by the Sullivan study. In the latter scenario, pre-exposure to an attractive odor won’t break the positive valence. We have tested perinatal exposure to attractive odors. As suggested by Reviewer 1, we have used TMA, which has been characterized as attractive in past studies. In our hand, TMA is less attractive than the compound odors from mouse urine or peanut butter. We also have tested exposure using peanut butter odor. In both cases, perinatal exposure extinguished innate attraction in the adults for the odor exposed, but not the un-exposed odor. This result shows that odor exposure breaks the valence association rather than counterbalance it. The data are presented in the updated Figure 2.

3. The results of the early PEA exposure experiments are intriguing, but further work is necessary to interpret the conclusions from these experiments. First, the authors use a concentration of 1:100 when exposure to μM concentrations (10,000 times less than what was used here) has been shown to suppress Taar4 transcript levels (Dewan 2018). Furthermore, the behavioral results in Figure 2 currently phenocopy the loss of activity in the Kir2.1 animals and the effects in TAAR4-KO animals (Dewan 2013), so it would be good to verify that the manipulations in the odor exposure experiments are indeed elevating activity levels. Therefore, the authors should monitor in more detail the physiological consequences of their early exposure paradigm (e.g. using the methods described in Figure 1B-C) and specifically assess that the high concentrations of odorants used are not reducing the OR levels in TAAR4-expressing neurons or altering the function of these neurons.

We thank the reviewer for raising this point and apologize for not making clear the rationale of using a high concentration for the odor exposure experiment. Dewan et al. (2013) showed that PEA elicited aversion from 0.005% to100% dilutions. In our study, the concentration for odor exposure was adjusted to how the animals access the odors. For the behavioral test in adults, the animals actively poke the odor port to smell the odor. PEA was tested at 1E-4 dilution, at the lower end of the Dewan et al. study. For early postnatal exposure, the pups were not mobile, and they would not have access to the odor source in the center of the home cage sealed in a lid with small holes. To reach a similar concentration at the pup, a higher concentration in the vial is warranted. We have measured the distribution of odor concentration as a function of distance to the odor vial. At the steady state, the concentration of PEA at 5cm from the source is about 5% of that in the vial (1E-2 dilution). Thus, the pups are estimated to be exposed to PEA at a concentration equivalent to 5E-4 dilution. This measurement is now included in Figure 2—figure supplement 1.

We are not able to record directly from the TAAR4 expressing cells because we do not have the GFP-tagged TAAR4 mouse line. Our attempt to stain TAAR4 receptor in the exposed animals was hampered by the problem with the antibodies. From the transcriptomic analyses, we find that TAAR4 expression was consistently reduced in PEA treated animals, even though the change is not considered statistically significant after adjusting for multiple comparison. This result is consistent with the Dewan et al. 2018 data and indicated that PEA did activate the TAAR4-expressing neurons. This data is presented in Figure 6G and Figure 6—figure supplement 1.

4. As the authors noted, it is surprising that surprising that the Kir2.1-off mice do not have major defects in odor detection, discrimination and association. Does this apply to the innate odorants (which can generally be reassigned meaning through reward): is detection and or discrimination impaired for any the innate odorants? Likewise, is the detection of PEA impaired after the early exposure paradigm?

We have performed experiments to examine the detection threshold of PEA in Kir-off and PEA-exposed mice. We also have examined cross habituation between PEA and eugenol. The results indicate that there is no difference in PEA detection and discrimination. The data are presented in the updated Figure 4—figure supplement.

5. It would be helpful if the authors could discuss their findings in the context of prior work studying the consequences of odor exposures on olfactory system plasticity and OB glomeruli. As an example, other odor exposure paradigms have resulted in increases in glomerular volume, changes in OSN numbers, or changes in OSN neurotransmitter release, among other effects (see e.g. Kass 2013, Bhattarai 2020, Jones 2008, Todrank 2010, Cadiou 2014, Ibarra-Soria 2017, Xu 2016). The results described in this manuscript differ from some of these; therefore, a more comprehensive discussion of prior work could be useful to contextualize and contrast the findings in this manuscript. There is also prior work on OSN-M/C targeting (e.g. Inoue 2018) and M/C dendritic stability (e.g. Mizrahi and Katz 2003).

We appreciate this suggestion and have revised the Discussion with a more contextualized discussion on the two topics. Additional citations have been included.

6. The introduction seeks to define differences between innate and learned behaviors, and map this onto predicted differences between circuits; many of the statements that end up being made are both overly didactic and highly debatable. To take just a single example on page 4 – "circuits that underlie innate responses are thought to be insensitive to sensory experiences." Obviously both worm chemoattraction and fly courtship are innate, hardwired and deeply plastic in response to sensory experience. The first three paragraphs have this quality (e.g., innate behaviors are not the same as fixed action patterns, it is not at all required that circuits that support innate behaviors be insulated from other types of information, what about retinal waves, etc). My advice is pretty simple – just say that we think of behaviors as innate because in adult animals we observe these behaviors and animals execute them without apparent prior experience or training. What this paper does is to go back and ask whether these innate behaviors actually depend upon early experience or neural activity; if so, that has implications for possible circuit mechanisms. Simple will clarify here.

We appreciate this suggestion and have revised the Introduction to make it more succinct.

Reviewer #3:The work by Qiu et al. describes experiments using transgenic mice in which spiking is selectively downregulated in OMP-expressing olfactory sensory neurons (OSNs) during the pre-weaning period. The authors find that innate attractive and aversive behavioral responses are lacking in these mice in adulthood. They also show that tonic presentation of an innately aversive odorant during the early postnatal period causes the loss of aversion to that odorant later in life. Both sets of behavioral results appear to correlate with changes in glomerular convergence and the activation of downstream brain regions associated with olfactory valence.The strength of this work is that it supports a critical role for early olfactory experience in future assignment of odor valence. Though many of the mechanisms underlying this phenomenon remain unknown/unexplored, this represents a compelling set of data and an important topic in the field. There are some notable limitations to these experiments that, if addressed in text or with inclusion of additional data or analysis, would improve confidence in the conclusions.1) In the Discussion (pp 23-24), the authors state: "Our results demonstrate the importance of single glomerulus axon convergence in providing a substrate for innate odor recognition." This is implying a causative role for the glomerular convergence in the observed changes in valence. This is too strong. These events are correlated, but it does not necessarily follow that the lack of glomerular convergence causes loss of innate valence (while supporting odorant discrimination).

We have revised the Discussion to provide more nuanced conclusions.

2) The results in Figure 4 are used to support the claim that DOX+ mice are still able to perform odor discrimination in adulthood. However, these experiments are all on neutral odors even though the central conclusion of the manuscript relates to odors with innate valence. It would improve this core conclusion if the manuscript included data on olfactory discrimination using odors with innate valence. For example, it would be useful to know whether DOX+ adult mice can discriminate PEA (which no longer causes innate aversion for these mice) from neutral or attractive odors.

We have performed experiments to examine the detection threshold of PEA and used cross habituation to examine odor discrimination in Kir-off and PEA-exposed mice. The results indicate that there is no difference in PEA detection and discrimination. The data are presented in the updated Figure 4—figure supplement.

3) The experiments present an interesting dichotomy: selectively silencing all OMP+ OSNs or constitutively activating OSNs that sense PEA both result in a loss of innate valence in adulthood. This suggests that the pattern of early postnatal activity in OSNs influences the animal's capacity to express an innate behavioral response to odorants later in life. This is exciting, but begs the question: does chronic postnatal exposure to an innately attractive odor (e.g. peanut oil) result in the same effect? If so, this might provide support for a unified mechanism of valence disruption (e.g. Figure 8). If not, this would still be interesting and would add depth to the study. At a minimum this should be discussed.

We appreciate this suggestion. We have conducted experiments for attractive odors, including the urinary compound TMA, and food odor (peanut butter). Early postnatal exposure in both cases diminished innate preference in an odor-specific manner, indicating that this is a general phenomenon. The data is presented in the updated Figure 2.

4) The analysis of spiking following DOX treatment in OMP-tTA x tetO-Kir2 mice seems to indicate a subtle, but potentially important change in the pattern of DOX+ OSN spiking towards burst-like firing. The ISI distribution appears to be skewed towards lower ISIs, but the analysis related to Figure 1B (a Student's t-test) indicates a lack of statistical significance. A Student's t-test does not seem to be an appropriate test for these non-normally distributed data. Also, there remain fairly meager data related to the transition from near-silence of OSNs in the DOX- case to the DOX+ post-weaning conditions. If this period involves transitions from near-silence to widespread bursting in all OMP+ neurons prior to a mildly bursty adult steady-state, this would seem to be important.

We appreciate this comment. Kir2.1 expression severely dampens spontaneous firing and odor evoked responses. Both may have an impact on odor-guided behaviors and brain activation. Therefore, the recording experiments were primarily designed to ensure that in the Kir2.1-off mice the neurons were able to fire action potentials and respond to odor stimulation. Neurons expressing different odorant receptors have different patterns of spontaneous activity. Although we wish to know whether all OMP+ neurons have been restored to their characteristic firing patterns, the heterogeneity of the OSN neurons makes it impractical to use electrophysiology to monitor the spiking activities. That is why we focused on the M71-GFP neurons to record spiking activities. Since our behavioral and brain activation studies were conducted at the adult stage, we only analyzed the response at this stage.

The reviewer is correct that the ISIs does not follow normal distribution. Theoretically they are thought to follow a Poisson process, but the data suggest that they follow a lognormal distribution. We have updated our plot and performed ANOVA on the mean log (ISI) values, which shows that the two distribution as not different (*p*=0.45).